# Genome-wide association study of REM sleep behavior disorder identifies polygenic risk and brain expression effects

Lynne Krohn[1,2], Karl Heilbron[3], Cornelis Blauwendraat[4], Regina H. Reynolds[5,6], Eric Yu[1,2], Konstantin Senkevich[1,2], Uladzislau Rudakou[1,2], Mehrdad A. Estiar[1,2], Emil K. Gustavsson[6,7], Kajsa Brolin[8], Jennifer A. Ruskey[2], Kathryn Freeman[2], Farnaz Asayesh[2], Ruth Chia[4], Isabelle Arnulf[9], Michele T. M. Hu[10,11], Jacques Y. Montplaisir[12,13], Jean-François Gagnon[12,14], Alex Desautels[12,15], Yves Dauvilliers[16], Gian Luigi Gigli[17], Mariarosaria Valente[17,18], Francesco Janes[17], Andrea Bernardini[17], Birgit Högl[19], Ambra Stefani[19], Abubaker Ibrahim[19], Karel Šonka[20], David Kemlink[20], Wolfgang Oertel[21], Annette Janzen[21], Giuseppe Plazzi[22,23], Francesco Biscarini[24], Elena Antelmi[23,25], Michela Figorilli[26], Monica Puligheddu[26], Brit Mollenhauer[27,28], Claudia Trenkwalder[27,28], Friederike Sixel-Döring[21,27], Valérie Cochen De Cock[29,30], Christelle Charley Monaca[31], Anna Heidbreder[32], Luigi Ferini-Strambi[33], Femke Dijkstra[34,35,36], Mineke Viaene[34,35], Beatriz Abril[37], Bradley F. Boeve[38], 23andMe Research Team*, Sonja W. Scholz[39,40], Mina Ryten[6,7], Sara Bandres-Ciga[4], Alastair Noyce[41,42], Paul Cannon[3], Lasse Pihlstrøm[43], Mike A. Nalls[44,45], Andrew B. Singleton[4,45], Guy A. Rouleau[1,2,46], Ronald B. Postuma[2,46] & Ziv Gan-Or[1,2,46] ✉

Rapid-eye movement (REM) sleep behavior disorder (RBD), enactment of dreams during REM sleep, is an early clinical symptom of alpha-synucleinopathies and defines a more severe subtype. The genetic background of RBD and its underlying mechanisms are not well understood. Here, we perform a genome-wide association study of RBD, identifying five RBD risk loci near *SNCA, GBA, TMEM175, INPP5F,* and *SCARB2*. Expression analyses highlight *SNCA-AS1* and potentially *SCARB2* differential expression in different brain regions in RBD, with *SNCA-AS1* further supported by colocalization analyses. Polygenic risk score, pathway analysis, and genetic correlations provide further insights into RBD genetics, highlighting RBD as a unique alpha-synucleinopathy subpopulation that will allow future early intervention.

Rapid-eye-movement (REM) sleep behavior disorder (RBD), defined as loss of muscle atonia and dream enactment during REM sleep, is one of the most unique conditions in neurology[1]. Isolated RBD (iRBD), defined as having RBD without other significant clinical neurological signs, is the only early highly predictive clinical marker for some neurodegenerative diseases. Over 80% of iRBD patients will convert within 10–15 years on average, to Parkinson's disease (PD), dementia with Lewy bodies (DLB), or in rare cases, multiple system atrophy (MSA)[2,3]. It is still unclear whether the remaining iRBD patients who did not convert at long follow-up will eventually convert, and pathological

A full list of affiliations appears at the end of the paper. *A list of authors and their affiliations appears at the end of the paper. ✉e-mail: ziv.gan-or@mcgill.ca

and imaging studies of this specific population are warranted. Since PD, DLB, and MSA are all characterized by accumulation of the protein alpha-synuclein, iRBD is considered a prodromal alpha-synucleinopathy, which offers a unique opportunity to identify these conditions at a much earlier stage[4].

There is strong evidence that iRBD also represents distinct, more severe subtypes of alpha-synucleinopathies. Approximately 30–60% of PD patients have RBD, including both iRBD and RBD as a symptom occurring after PD diagnosis (symptomatic RBD, sRBD)[3]. In this manuscript, we will use "RBD" to refer to all instances of RBD regardless of at which stage symptoms present, and iRBD or sRBD to specify before or after overt neurodegeneration diagnosis, respectively. The presence of RBD is currently the strongest predictor for the development of dementia in PD[5] and is associated with more rapid progression of non-motor symptoms[3]. RBD is more frequent in DLB, found in ~50–80% of all cases, and is associated with increased severity of DLB symptoms and rapid deterioration[6]. MSA patients also have a high prevalence of RBD, estimated at 75–95%, 40% of which have iRBD. Those with iRBD may have more frequent autonomic onset of MSA, less frequent parkinsonism at MSA onset, and a more severe disease course[7]. Overall, RBD, and specifically iRBD, appears to represent a more malignant subtype of alpha-synucleinopathies.

Thus far, the genetics of RBD has only been studied through the candidate gene approach. To better understand RBD and early alpha-synucleinopathy genetics and potential mechanisms, we performed a genome-wide association study (GWAS) on 2843 cases and 139,636 controls. We further examined the biological implications of the nominated risk loci through pathway analysis, investigated variant effects on gene expression, and assessed the cumulative risk using polygenic risk score (PRS). Using the GWAS summary statistics, we studied the genetic relationship between RBD and the synucleinopathies to which it progresses, as well as other conditions and exposures of interest.

## Results

### Genome-wide association study identifies five RBD loci

To identify genetic risk loci across the genome associated with RBD, we performed a case–control GWAS of iRBD ($N$ cases = 1061, $N$ controls = 8386) and a case–control GWAS from 23andMe, Inc. using PD patients with probable RBD (PD+pRBD) and controls without PD or RBD ($N$ cases = 1782, $N$ controls = 131,250), meta-analyzed for a total of 2843 cases and 139,636 controls. We tested for systemic biases using the genomic inflation factor (lambda), LD-score regression, and QQ-plots (Supplementary Fig. 1), with satisfactory results (lambda = 1.06, lambda1000 = 1.01, LD intercept = −0.01). With LD-score regression, the liability-scale narrow-sense heritability of iRBD based on common variants is calculated at 12.3% (standard error = 0.07), similar to the recently reported 10.8% heritability for DLB[8].

We identified RBD-associated loci in *SCARB2* and *INPP5F*, and replicated known RBD associations near *SNCA*[9], *TMEM175*[10], and two variants in *GBA*, p.Glu326Lys and p.Asn370Ser[11,12] (Table 1 and Fig. 1). Individual LocusZoom plots can be found in Supplementary Fig. 2. These five loci have also been implicated in PD[13], however, the RBD-associated SNPs in *SNCA* and *SCARB2* are not in LD with the top PD-associated SNPs in these loci, and are thus considered independent. No secondary associations were identified by conditional-joint analysis; notably, the PD variant is not significant at the *SCARB2* locus (Supplementary Fig. 3). The *SNCA* locus structure in RBD has been extensively studied before, with a potential secondary hit that is below genome-wide statistical significance[9]. Additionally, PD or DLB-associated SNPs in notable GWAS loci, such as *MAPT* (rs62053943), *LRRK2* (rs34637584), *BIN1* (rs6733839) and *APOE* (rs769449)[8,14] are not associated with RBD at this sample size, which had sufficient power (>80%) to detect the effect sizes seen in PD and DLB. This suggests that RBD-associated synucleinopathy may have only partially overlapping genetic background with overall PD and DLB.

**Table 1 | Independent RBD risk loci nominated by GWAS meta-analysis**

| Meta-analysis | | | | | | | | | | PD+pRBD | | | iRBD | | |
|---|---|---|---|---|---|---|---|---|---|---|---|---|---|---|---|
| Position (hg19) | SNP | Closest gene | Eff allele | Ref allele | EAF | OR | 95% CI | p | Het I² (%) | OR | 95% CI | p | OR | 95% CI | p |
| 4:90757272 | rs3756059 | SNCA | A | G | 0.50 | 1.26 | 1.19–1.33 | 3.02E-16 | 94.4 | 1.16 | 1.08–1.24 | 2.67E-05 | 1.49 | 1.36–1.64 | 2.42E-16 |
| 1:155205378 | rs12752133 | GBA | T | C | 0.01 | 2.09 | 1.73–2.54 | 4.87E-14 | 0 | 2.06 | 1.64–2.59 | 1.65E-08 | 2.18 | 1.53–3.11 | 1.78E-05 |
| 1:155205634 | rs76763715 | GBA | C | T | 0.004 | 2.84 | 2.06–3.92 | 1.68E-10 | 0 | 2.74 | 1.9–3.94 | 1.35E-06 | 3.25 | 1.65–6.41 | 6.79E-04 |
| 4:951947 | rs34311866 | TMEM175 | C | T | 0.19 | 1.22 | 1.14–1.31 | 4.41E-09 | 0 | 1.2 | 1.1–1.3 | 2.45E-05 | 1.28 | 1.14–1.44 | 3.99E-05 |
| 10:121536327 | rs117896735 | INPP5F/BAG3 | A | G | 0.02 | 1.8 | 1.48–2.19 | 4.70E-09 | 0 | 1.88 | 1.5–2.36 | 4.04E-07 | 1.57 | 1.05–2.34 | 2.68E-02 |
| 4:77132634 | rs7697073 | SCARB2 | T | C | 0.34 | 1.18 | 1.11–1.25 | 2.21E-08 | 0 | 1.16 | 1.08–1.25 | 2.71E-05 | 1.2 | 1.09–1.33 | 2.03E-04 |

We identified 6 independent variants significantly associated with RBD through genome-wide association study (GWAS), performed as repeated logistic regression across the genome, adjusted for age, sex, and principal components. Loci are considered significant if their two-sided $p$-value is less than the standard GWAS multiple testing corrected threshold ($p$ < 5E-08). The high heterogeneity found in the SNCA locus could be attributed to the stronger effect size in iRBD (OR = 1.49) compared to PD+pRBD (OR = 1.16). The two GBA associations, representing the p.Glu326Lys and p.Asn370Ser variants, are not in LD, as they are known from Gaucher's disease studies to reside on different alleles.

*PD* Parkinson's disease, *pRBD* probable REM sleep behavior disorder, *iRBD* idiopathic REM sleep behavior disorder, *SNP* single-nucleotide polymorphism, *Eff* effect, *Ref* reference, *EAF* effect allele frequency, *OR* odds ratio, *CI* confidence interval, *Het* heterogeneity.

**a**

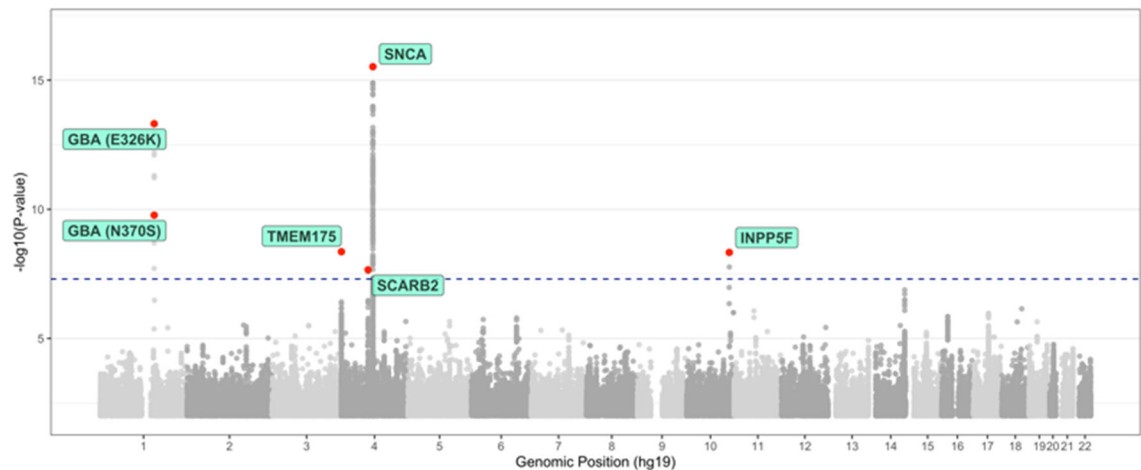

**b**

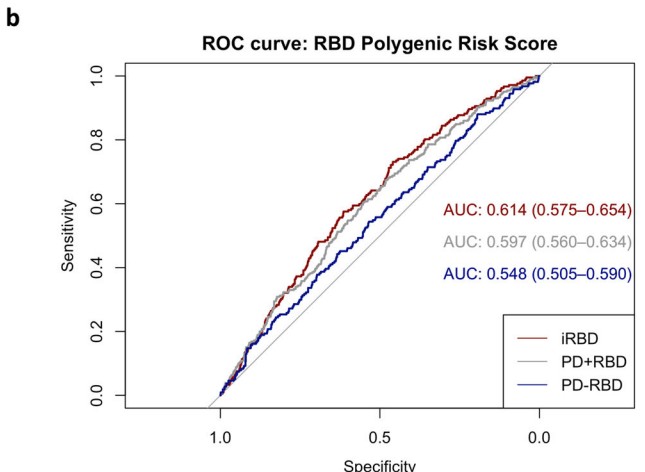

**c**

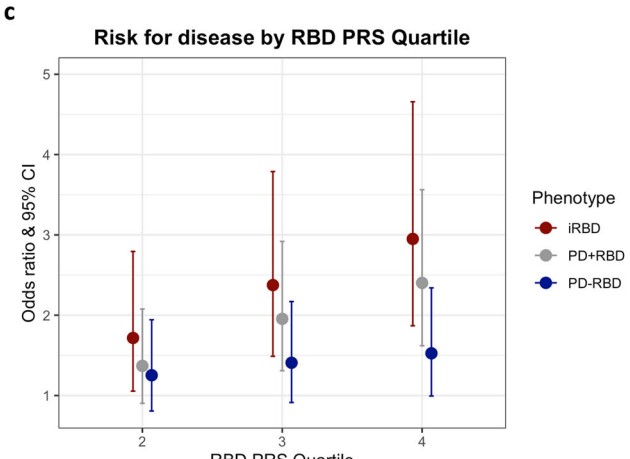

**Fig. 1 | Summary of GWAS findings in the RBD meta-analysis. Manhattan plot.**
*ROC* receiver operating characteristic, *iRBD* idiopathic REM sleep behavior disorder, *PRS* polygenic risk score, *PD* Parkinson's disease, *pRBD* probable RBD. **a** The Manhattan plot highlights the 6 GWAS-nominated loci after meta-analysis. GWAS was performed as repeated logistic regression across the genome, adjusted for age, sex, and principal components. Each point represents the log adjusted *p*-value at each genomic site. A locus was considered significant if the two-sided *p*-value was less than the corrected GWAS-significant *p*-value threshold of 5E-08, visualized in this plot with the dashed line. The points in red show the top variant at that locus, as well as any secondary independent associations. Predictive power of RBD polygenic risk score. Polygenic risk scores for RBD were calculated using FDR-corrected GWAS variants (*N* SNPs = 47) in 3 cohorts: idiopathic RBD, PD+pRBD, and PD−pRBD, each with controls. **b** The predictive power of the PRS in each cohort was assessed with area under the curve (AUC) and 95% confidence intervals. **c** The PRS for each cohort were divided in quartiles and analyzed against case status with logistic regression (*N* iRBD = 212 with *N* controls = 1265; *N* PD+pRBD = 285 with *N* controls = 907; *N* PD−pRBD = 217 with *N* controls = 907). The odds ratios and 95% confidence intervals (the odds ratio +/− 1.96*standard error) are visualized here as compared to the lowest quartile (the lowest 25% of scores).

Additionally, we investigated common variants in loci where rare variants are associated with reduced RBD risk: *BST1* and *LAMP3*[15]. Intronic *BST1* SNP rs4389574 (MAF = 0.45) shows a potential protective effect in the RBD meta-analysis, without confidence (beta = −0.07, se = 0.03 m *p* = 0.01). Intronic *LAMP3* SNP rs3772714 (MAF = 0.14) also shows a potentially protective effect, again without confidence at GWAS-corrected significance (beta = −0.14, se = 0.04, *p* = 0.001). Linkage disequilibrium (LD) is difficult to assess between common and rare variants; in this case, LDlink[16] shows the minor alleles of the RBD rare variants in *BST1* (rs6840615) and *LAMP3* (rs56682988) correlate with the minor alleles for the common variants reported above, however in both cases only one instance of the rare variant was identified.

### RBD polygenic risk scores better predict RBD case status than PD without RBD
Next, we examined whether RBD-specific PRS can distinctly identify RBD as opposed to PD without RBD, using an RBD polygenic risk profile containing independent risk variants reaching FDR-corrected

significance in this meta-analysis (*N* = 47, Supplementary Data 1). In iRBD, the PRS can differentiate between iRBD cases and controls with an area under the curve (AUC) of 0.61 (95% CI = 0.58–0.65, Fig. 1b), on par with recent PD PRS performance (AUC = 0.62)[17]. In PD+pRBD, predictive performance was similar to iRBD with an AUC of 0.60 (95% CI = 0.56–0.63), compared to decreased predictive power in PD−pRBD with an AUC = 0.55 (95% CI = 0.51–0.59, Fig. 1b). Using DeLong's test to assess true difference between these AUC, PD+pRBD and PD−pRBD do not differ with statistical significance (*p* = 0.088), however, the PRS performs better in iRBD than PD−pRBD (*p* = 0.024). When comparing PD+pRBD to PD−pRBD, the RBD PRS is not a strong predictor (AUC = 0.55, 95% CI 0.52–0.59). However, while we argue this RBD PRS is enriched for RBD loci, it does not capture other key differences we may expect to see in PD +/− RBD, such as *LRRK2* and *MAPT* variants. Further statistical investigation into PD +/− RBD genetic differences is warranted. Results are comparable when excluding rare *GBA* variant p.Asn370Ser from the polygenic risk profile, with only the iRBD AUC decreasing from 0.61 to 0.60 (95% CI 0.56–0.64), but the

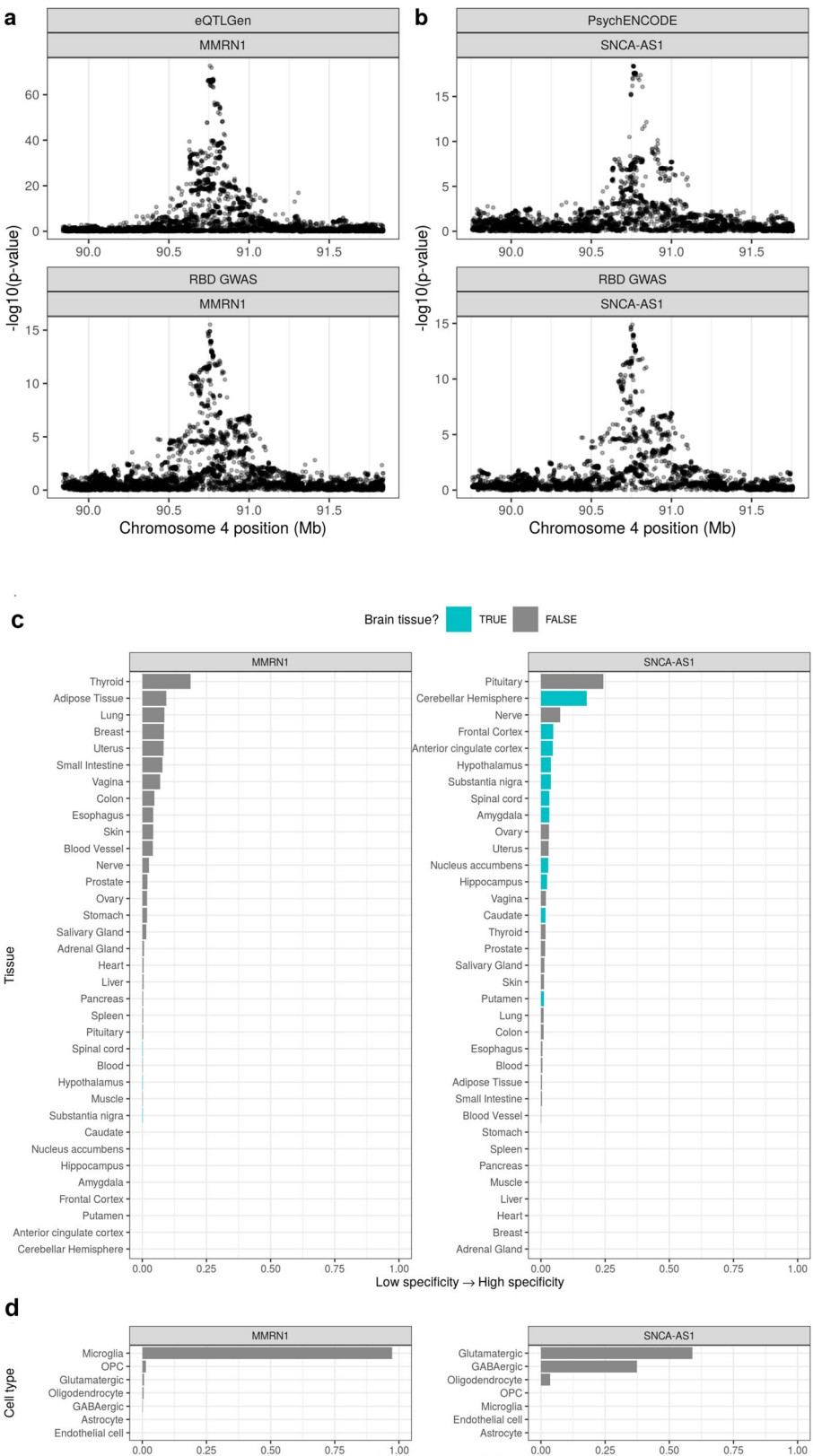

difference between iRBD and PD−pRBD AUC becomes no longer significant ($p = 0.06$).

We then divided the RBD PRS into quartiles in each cohort and performed logistic regression against the phenotype (Fig. 1c). In iRBD, those in the top quartile for RBD PRS were 2.9 times more likely to have RBD (95% CI = 1.87–4.66, $p = 3.5E{-}06$), while in PD+pRBD those in the

top quartile were 2.4 times more likely to have PD+pRBD (95% CI = 1.62–3.56, $p = 1.0E{-}05$). The RBD PRS is not significantly associated with risk for PD−pRBD; those in the top quartile were 1.53 times more likely to have PD−pRBD, without statistical significance (95% CI = 0.99–2.34, uncorrected $p = 0.053$, Fig. 1c). The lack of RBD PRS predictive power in PD−pRBD suggests that we are likely tagging RBD-

**Fig. 2 | Regional association plots for eQTL and RBD GWAS colocalizations and tissue and cell-type specificity of *MMRN1* and *SNCA-AS1*.** *RBD* REM sleep behavior disorder, *GWAS* genome-wide association study. Regional association plots for eQTL (upper pane) and RBD GWAS association signals (lower pane) in the regions surrounding **a** *SNCA-AS1* (colocalization PPH4 = 0.89) and **b** *MMRN1* (colocalization PPH4 = 0.86). eQTLs are derived from **a** PsychENCODE's analysis of adult brain tissue from 1387 individuals or **b** the eQTLGen meta-analysis of 31,684 blood samples from 37 cohorts. In **a** and **b**, the *x*-axis denotes chromosomal position in hg19, and the *y*-axis indicates association *p*-values from across-locus logistic (RBD GWAS) or linear (eQTL) regression on a −log₁₀ scale. Plot of *SNCA-AS1* and *MMRN1* specificity in **c** 35 human tissues (GTEx dataset) and **d** 7 broad categories of cell type derived from human middle temporal gyrus (AIBS dataset). Specificity represents the proportion of a gene's total expression attributable to one cell type/tissue, with a value of 0 meaning a gene is not expressed in that cell type/tissue and a value of 1 meaning that a gene is only expressed in that cell type/tissue. In **c** tissues are colored by whether they belong to the brain. In **c** and **d**, tissues and cell types have been ordered by specificity from high to low.

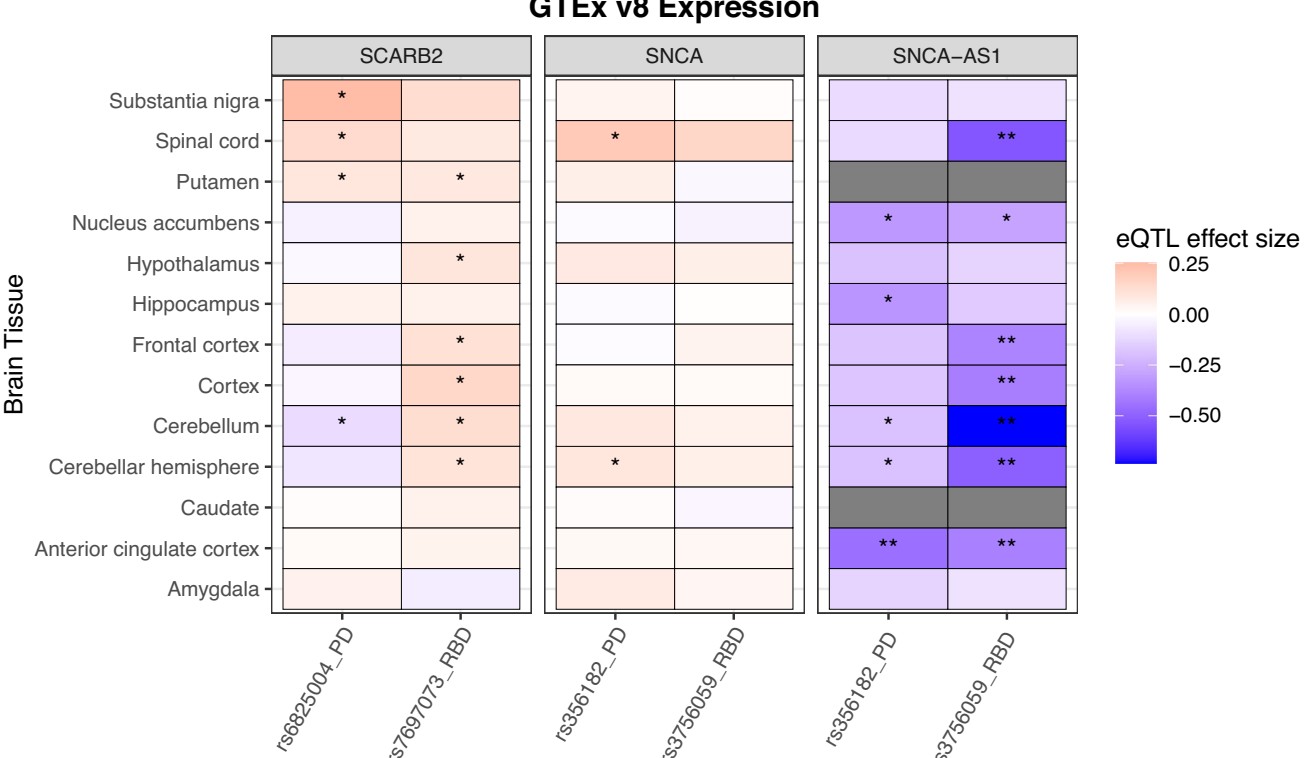

**Fig. 3 | eQTL data from GTEx version 8 for RBD and PD top variants in differing loci.** *GTEx* Genotype-Tissue Expression Consortium, *v8* version 8, *RBD* REM sleep behavior disorder, *PD* Parkinson's disease, *eQTL* expression quantitative trait loci. All data was extracted from the GTEx online portal (https://www.gtexportal.org/). The effect sizes represent the slope of linear regression on normalized gene expression data versus the genotype status using single-tissue eQTL analysis, performed by the GTEx consortium. Nominal associations are indicated with* (*p* < 0.05) while FDR-corrected significant associations are indicated with**. FDR correction *q*-values were calculated using beta distribution-adjusted empiracle p-values, derived from adaptive permutations during eQTL mapping. Dark gray indicates missing data in these tissues. The RBD *SNCA* variant correlates most strongly with decreased *SNCA-AS1* expression in the cerebellum (uncorrected *p* = 9.9e-19) cerebellar hemisphere (*p* = 3.1e-08), cortex (*p* = 1.9e-05), frontal cortex (*p* = 6.3e-05), and anterior cingulate cortex (*p* = 2.6e-04). The PD variant only significantly correlates with decreased expression in the anterior cingulate cortex (*p* = 2.5e-05).

specific loci. RBD PRS was not associated with changes in RBD AAO, PD AAO, or rate of conversion from RBD to overt neurodegeneration.

## Colocalization analyses demonstrate tissue and cell-specific differential effects of RBD-associated variants

We further performed colocalization analyses to determine whether risk variants for RBD are also associated with gene expression in the whole brain or blood. Expression quantitative trait loci (eQTLs) were obtained from PsychENCODE[18] and eQTLGen[19], large human brain and blood datasets, respectively. In brain, we found strong evidence for colocalization in the *SNCA* locus with *SNCA*antisense-1 (*SNCA-AS1*) expression (posterior probability of hypothesis 4, PPH4 = 0.89; Fig. 2a and Supplementary Data 2). SNPs in the region surrounding *SNCA-AS1* tended to show an inverse relationship between RBD risk and *SNCA-AS1* expression, suggesting that reduced RBD risk is associated with increased *SNCA-AS1* expression (Supplementary Fig. 4), which in turn may be associated with reduced alpha-synuclein protein level.

In blood, we found evidence of colocalizations in the *SNCA* locus with *MMRN1* expression (*MMRN1*, PPH4 = 0.86; Fig. 2b and Supplementary Data 2). Sensitivity analyses confirmed that *SNCA-AS1* and *MMRN1* colocalizations were robust to changes in the prior probability of a variant associating with both traits (i.e., *p₁₂* prior, Supplementary Figs. 5 and 6).

As both the *SNCA-AS1* and *MMRN1* colocalizations were observed at the same RBD risk locus, but in different tissues, we hypothesized that this may be due to tissue-specific regulation of *SNCA-AS1* and *MMRN1* expression. Using specificity, a measure of the proportion of a gene's total expression attributable to one tissue or cell type, we explored the tissue- and cell-type-specific patterns of *SNCA-AS1* and *MMRN1* expression in (i) human bulk-tissue RNA-sequencing from GTEx consortium[20] and (ii) human single-nucleus RNA-sequencing of the medial temporal gyrus from the Allen Institute for Brain Science (AIBS; 7 cell types)[21]. At the tissue level, *SNCA-AS1* expression was predominantly brain-specific, while *MMRN1* expression was most specific to thyroid, adipose and lung tissues and least specific to brain

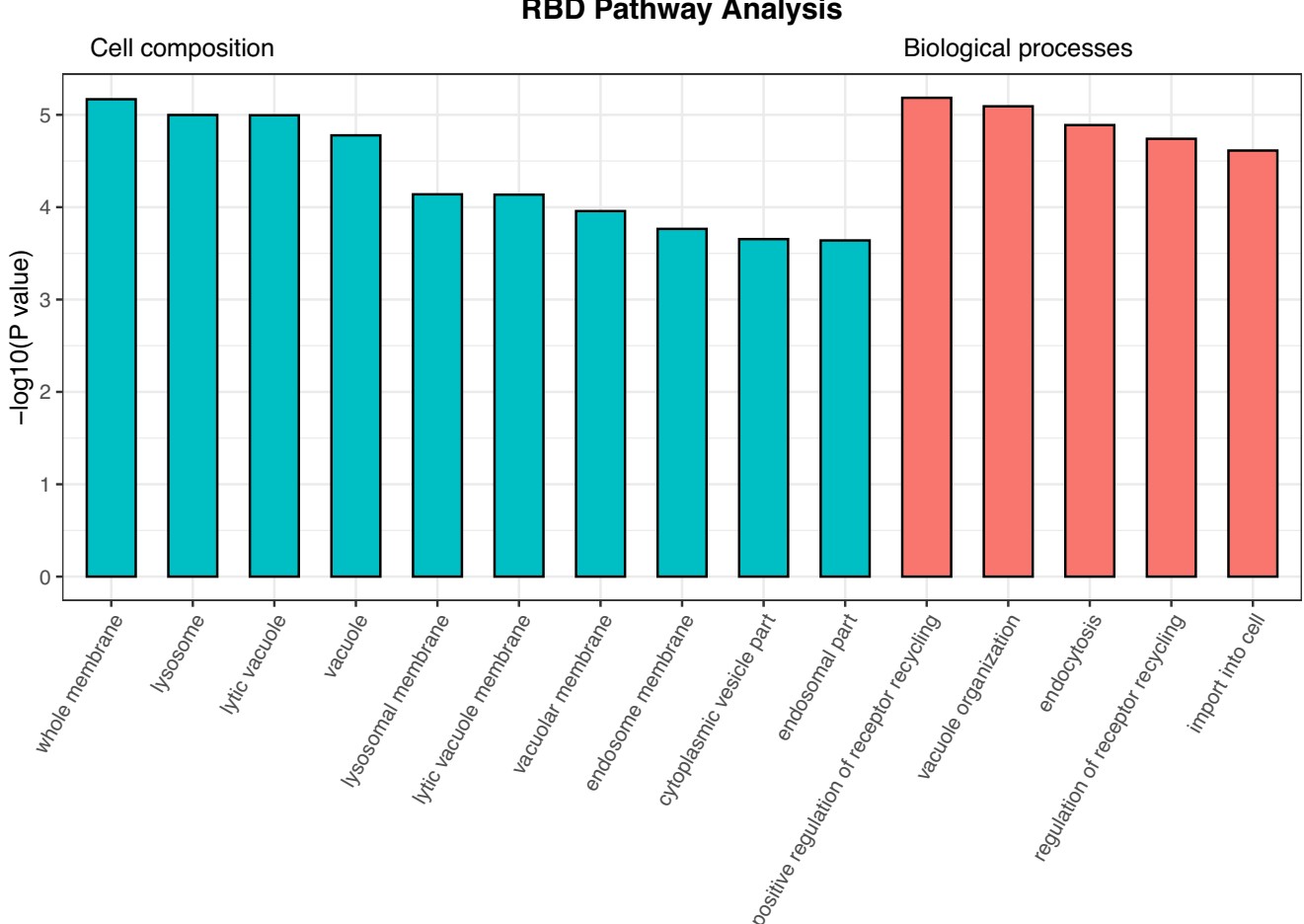

**Fig. 4 | Pathways associated with genes nominated by RBD meta-analysis.** *RBD* REM sleep behavior disorder. We used WebGestalt (http://www.webgestalt.org/) to perform gene-set enrichment analysis in cellular components and biological processes. Enrichment scores are calculated similar to Kolmogorov–Smirnov statistics; they indicate whether a group of genes is over- or under-represented in a list of processes. Two-sided *p*-values are calculated by comparison to the enrichment scores null distribution, produced with phenotypic permutation testing. Bars on this plot represent the unadjusted *p*-values for pathways nominated by gene-set enrichment analysis. All pictured pathways are significant after FDR multiple testing correction.

tissues (Fig. 2c and Supplementary Data 3). At the cellular level, *SNCA-AS1* demonstrated neuronal specificity, while *MMRN1* was specific to microglia (Fig. 2d and Supplementary Data 3). We further investigated whether these associations in CNS cell-type-specific eQTLs[22], but did not find evidence of colocalization with any RBD loci.

**Differential eQTL effects in different brain regions may shed light the independent associations of SNCA in RBD and PD**
*SNCA* and *SCARB2* are GWAS-nominated risk loci for both RBD and PD, however the associations are driven by independent variants. In the *SNCA* locus, rs3756059 (in the 5′ region) was associated with RBD in the current GWAS and recently reported in DLB[8]. In contrast, rs356182 (in the 3′ region), which is not in LD with rs3756059 ($R^2 = 0.17$, $D' = 0.56$), is the most significant GWAS signal for PD risk[13], yet showed no association with iRBD. Similarly, in the *SCARB2* locus, rs7697073 is associated with RBD and rs6825004 is associated with PD, and the two SNPs are not in LD ($R^2 = 0.06$, $D' = 0.26$). We therefore hypothesized that the different SNPs in these loci may be associated with differential expression patterns of their respective genes in different brain regions. To examine this hypothesis, we used the Genotype-Tissue Expression (GTEx) consortium v8 (https://www.gtexportal.org/home/)[23]. We examined the effects of these variants on the expression of *SCARB2*, *SNCA* and *SNCA-AS1*, since variants in *SNCA* have previously been linked to *SNCA-AS1* expression[8], and our colocalization analysis suggests the RBD locus is driven by *SNCA-AS1* expression. The eQTL effect size (ES) reported here is the slope of the linear regression of normalized expression data versus the genotype status using single-tissue eQTL analysis, performed by the GTEx consortium. A variant is considered associated with expression in a tissue based on its association with mRNA levels after FDR correction by GTEx, however in our case it does not indicate that this is the causal eQTL; it may be in LD with the causal SNP. Comparative results are visualized in Fig. 3.

In the *SNCA/SNCA-AS1* locus, the RBD risk variant rs3756059 is most strongly associated with increased expression for *SNCA-AS1* expression in multiple cortical regions (frontal cortex effect size, ES = −0.39, *p* = 6.3E-05; anterior cingulate cortex ES = −0.41, *p* = 2.6E-04), the cerebellum (ES = −0.74, *p* = 9.9E-19) and the spinal cord (ES = −0.54, *p* = 2.0E-05), all statistically significant. The PD variant rs356182 is only associated with *SNCA-AS1* expression in the anterior cingulate cortex (ES = −0.46, *p* = 2.5E-05). Although the observed direction of eQTL effect on *SNCA-AS1* expression is consistent between the RBD and PD variants, only the RBD variant is causally linked to *SNCA*-AS1 expression via colocalization. The differential strength and patterns across the brain regions between the PD and RBD variants are an intriguing field for follow-up investigation; in individual GTEx data, statistically comparing the mRNA levels data associated with the PD or RBD variant in each tissue could help clarify this observation.

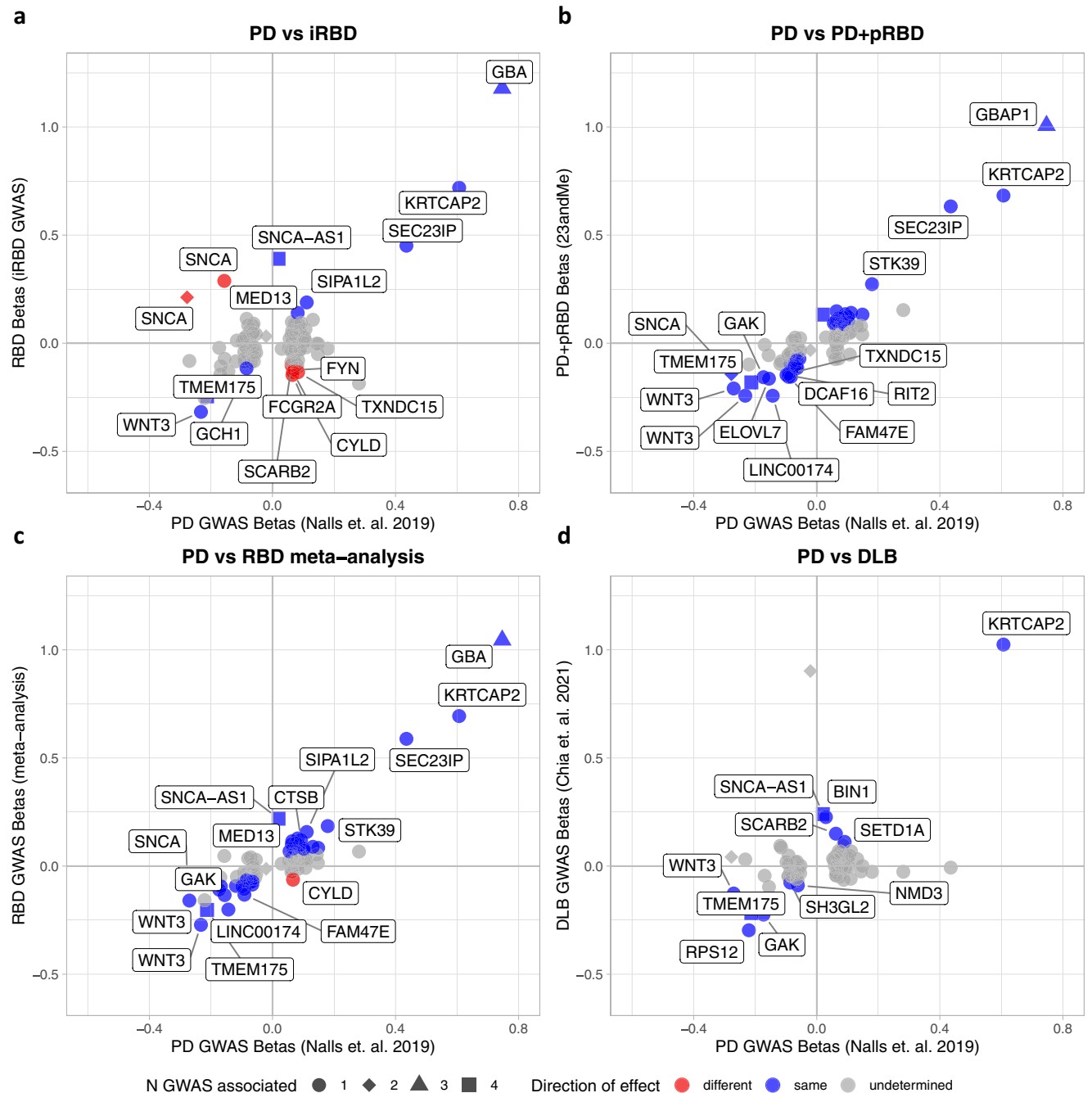

**Fig. 5 | Beta-beta plots comparing synucleinopathy GWAS summary statistics to the latest PD GWAS.** *PD* Parkinson's disease, *RBD* REM sleep behavior disorder, *GWAS* genome-wide association study, *pRBD* probable RBD, *DLB* dementia with Lewy bodies. We compare significance and direction of PD GWAS-nominated loci to this study's summary statistics for iRBD (**a**), PD+pRBD (**b**), the meta-analysis (**c**), and in the previously published DLB summary statistics (**d**). Colored points indicate variants with the same (blue) or opposite (red) direction of effect in both studies, with a nominally significant *p*-value (*p* < 0.05) in their respective genome-wide association studies (two-sided *p*-value derived from logistic regression across the genome). All test statistics for each cohort can be found in Supplementary Table 5. Gray points are those with undetermined direction (*p* > 0.05 and confidence intervals cross 0). The shapes of the points indicate the number of synucleinopathy GWAS where the locus reaches GWAS significance (counting PD, PD age at onset, DLB, and this RBD meta-analysis). Gene names indicate the closest gene to the represented variant.

The RBD risk variant at the *SCARB2* locus (rs7697073) is most strongly associated with increased expression in the cortex (ES = 0.15, *p* = 1.4E-03), while the *SCARB2* PD risk variant (rs6825004) is most strongly associated with substantia nigra increased expression (ES = 0.25, *p* = 4.6E-04). The PD variant may be associated with decreased *SCARB2* expression in the cerebellum, while the opposite is shown for the RBD variant. These *SCARB2* results must be taken with caution and examined further, as the eQTL associations are not statistically significant after multiple testing correction, and we did not find evidence of colocalization in the RBD *SCARB2* locus in brain or blood (Supplementary Figs. 7–9).

### Pathway analysis reveals potential role for the autophagy-lysosomal pathway in RBD pathogenesis

To examine whether specific pathways are enriched according to the RBD GWAS results, we performed pathway enrichment analysis using

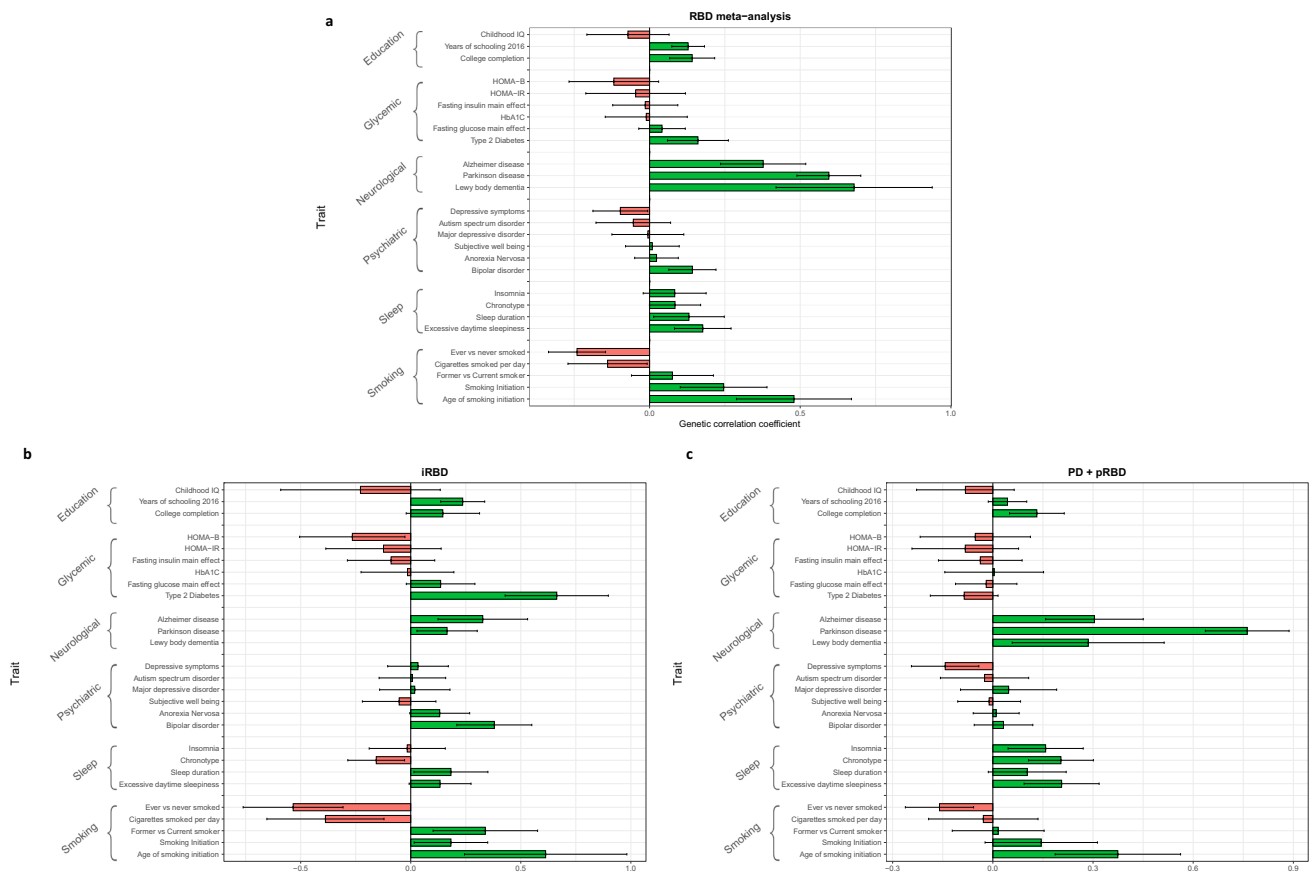

**Fig. 6 | Genetic correlation results.** *RBD* REM sleep behavior disorder, *PD* Parkinson's disease, *pRBD* probable REM sleep behavior disorder. Genetic correlation was calculated using LD-score regression for **a** the RBD meta-analysis (N cases = 2843 and N controls = 139,636), **b** isolated RBD (iRBD) alone (N cases = 1061 and N controls = 8386), and **c** PD+pRBD alone (N cases = 1782 and N controls = 131,250). The traits tested are organized in their general categories as labeled by LD Hub. The error bars represent the genetic correlation coefficient +/− the standard error, centered on the correlation coefficient.

the GWAS-nominated genes for cellular components and biological processes (Fig. 4 and Supplementary Data 4). The nominated cellular components include whole membrane (FDR corrected $p = 0.004$), lysosome ($p = 0.004$), and vacuole ($p = 0.005$). Biological processes include positive regulation of receptor recycling ($p = 0.037$), vacuole organization ($p = 0.037$), and endocytosis ($p = 0.039$). All these nominations suggest involvement of the autophagy-lysosomal pathway (ALP), a key mechanism for clearing alpha-synuclein[24,25].

### Comparison to PD GWAS loci reveals additional loci with potential distinct effects in RBD

We aimed to further examine how PD GWAS loci behave in RBD, given the differential associations we observed in the *SNCA* and *SCARB2* loci. We compiled a list of GWAS-nominated synucleinopathy variants (PD[13], DLB[8], PD AAO[26], and RBD) and compared the effects in PD GWAS summary statistics versus the summary statistics of this RBD GWAS (iRBD, PD+pRBD, and the meta-analysis) and the most recent DLB GWAS [8].

Figure 5a–c shows the similarities and differences of effect size and direction at these loci across RBD, PD and DLB. Notably, iRBD shows marked differences at some key PD loci (Fig. 5a), including *SNCA, CYLD*, and *FYN*. In these loci, the direction of effect in iRBD is opposite of that seen in PD, however without corrected significance in iRBD. The PD+pRBD cohort deviates from this pattern with 100% of loci showing the same direction of effect with PD (Fig. 5b). However, strong PD signals, such as *SNCA* 3' variants and *LRRK2*, are not significant in the PD+pRBD cohort despite sufficient power. Similarly, all DLB nominally significant loci share the same direction of effect with

PD (Fig. 5c). Yet, DLB also deviates from PD, as the *MAPT* (rs62053943), *LRRK2* (rs34637584), and *SNCA* 3' (rs356182) loci are not statistically significant despite sufficient power.

We repeated this comparison using PD AAO summary statistics, comparing the same set of variants to RBD and DLB GWAS statistics (Supplementary Fig. 10). As expected, most loci that increase risk for any of the synucleinopathies are associated with an earlier PD AAO, so we see a consistent "different" effect direction. However, in iRBD, we see a few with the opposite effect; notably, the PD *SCARB2* variant (PD beta = 0.06, $p = 1.17e{-}09$) is nominally associated with an earlier PD AAO (beta = −0.28, $p = 0.028$), but decreased risk for RBD (beta = −0.10, $p = 0.045$). Similarly, the top PD variant rs356182 is concordant in the PD and PD AAO summary statistics but not in iRBD, a distinction, which was noted in an earlier study of *SNCA* in RBD[9]. The allele associated with increased risk for PD (beta = 0.28, $p = 3.9e{-}154$) is associated with earlier PD AAO (beta = −0.67, $p = 4.6e{-}08$), but shows a potential for a protective effect in iRBD (beta = −0.23, $p = 1.5e{-}04$). All statistics are detailed in Supplementary Data 5.

### LD-score regression reflects potential differences between iRBD and PD+pRBD

We used LD-score regression to examine the genetic correlation between RBD and relevant traits and exposures (Fig. 6a–c). Although iRBD and PD+pRBD are positively correlated with nominal significance (rg = 0.56, se = 0.24, $p = 0.02$), the two cohorts behave differently when it comes to other alpha-synucleinopathies. PD+pRBD is strongly correlated with PD (rg = 0.76, se = 0.13, $p = 1.2E{-}09$), yet iRBD is not (rg = 0.17, se = 0.14, $p = 0.23$). PD+pRBD is not correlated with DLB (rg =

0.29, se = 0.23, p = 0.21), yet the meta-analysis of iRBD and PD+pRBD is positively correlated with DLB (rg = 0.68, se = 0.26, p = 0.009, not significant after multiple testing correction). Therefore, this possible association is likely driven by iRBD, although we could not accurately measure genetic correlation between iRBD and DLB due to high variability causing the correlation estimate to be out of bounds (rg = 1.28, se = 0.56, p = 0.02), suggesting we are underpowered to detect a true effect. Of note, we chose to analyze the latest DLB GWAS, which includes 150 PDD cases in the replication phase. This study is the largest to date and has consistent findings with the smaller 2018 study with only DLB cases[14]. This difference between iRBD and PD+pRBD is quite pronounced when examining genetic correlation between the synucleinopathies for only PD, DLB, and RBD GWAS loci +/− 500 kb. PD +pRBD is correlated with PD (rg = 0.76, se = 0.08, p = 1.2E-19) with uncertain results for DLB (rg = 1.31, se = 2.03, p = 0.52), and iRBD is not correlated with PD (rg = 0.19, se = 0.13, p = 0.15) or DLB, although with a potentially high correlation coefficient with DLB (rg = 0.91, se = 0.56, p = 0.17) with low confidence. Additionally, iRBD is potentially genetically correlated with Type II Diabetes (rg = 0.66, se = 0.23, p = 0.0047) without significance after multiple testing correction, while PD+pRBD is not (rg = −0.09, se = 0.10, p = 0.40). Interestingly, PD+pRBD may be correlated with Alzheimer's disease (rg = 0.30, se = 0.15, p = 0.04), again without confidence, which we do not see in iRBD at this sample size, although the correlation coefficients are similar (rg = 0.33, se = 0.20, p = 0.11). Those with Type II Diabetes are at increased risk for Alzheimer's[27] and the two conditions share genetic risk architecture[28]. All RBD cohorts show similarities to PD with potential genetic correlations with less smoking, more education, and excessive daytime sleepiness, yet without significance after multiple testing correction (Fig. 6a–c and Supplementary Data 6).

## Discussion

In this GWAS of RBD, we identified six RBD-associated loci in five genomic regions: two loci near *SCARB2* and *INPP5F* and three previously reported loci near *SNCA*, *GBA* and *TMEM175*. Two of the loci, *SNCA* and *SCARB2*, have different and independent variants associated with RBD than those associated with PD. Our sample size had sufficient power (>80%) to detect variants previously reported in PD and DLB at the GWAS significance level, namely, *MAPT* (rs62053943), *LRRK2* (rs34637584), *BIN1* (rs6733839) and *APOE* (rs769449)[8,14], yet there was no association. This does not provide a definite proof that these variants are not associated with RBD, yet it further suggests a genetic background in the RBD subtype that is only partially overlapping with those of PD and DLB as a whole. Colocalization analyses suggest that the *SNCA* variants are associated with differential expression of *SNCA-AS1* in different brain regions, tissues, and cell types. We further show that RBD-specific PRS better predicts RBD case status than PD−pRBD when compared to controls. Pathway analysis and comparisons between this RBD meta-analysis and previous PD and DLB GWAS show specificity to the autophagy-lysosomal pathway (ALP) in risk for iRBD and RBD-associated PD, whereas that specificity is not noted in similarly powered PD or DLB GWAS. Based on these results, we hypothesize that RBD is an indicator of a specific synucleinopathy subtype, which is characterized by ALP dysfunction and cognitive decline, with clinical and genetic distinctions, which should be considered in future drug development and clinical trials.

The differential association at the *SNCA* locus, when comparing PD and RBD, may provide a mechanistic hypothesis for gene expression-dependent regional vulnerability of different brain areas. The eQTL analysis shows that the top variant associated with RBD in the current GWAS, rs3756059, is associated with reduced expression of *SNCA-AS1* in several cortical regions (Fig. 3) while the top PD variant (rs356182) is not. Since *SNCA-AS1* is transcribed as an antisense RNA molecule, it might lead to reduced alpha-synuclein protein expression. If this hypothesis is correct, then levels of alpha-synuclein protein may

be increased in the cortical regions associated with reduced *SNCA-AS1* expression (Frontal Cortex), which could make these regions more vulnerable for neurodegeneration in carriers of the RBD-associated variant. This could explain some of the strong association of RBD with more rapid and more severe cognitive decline in PD[29,30], since the cortical brain regions associated with cognition may be more susceptible for neurodegeneration by this mechanism. Interestingly, PD patients without RBD are similar to controls when assessing cognition[30,31], and this RBD variant is not a strong risk locus in this PD−RBD population[9]. This locus is in LD with a secondary PD GWAS variant, rs7681154, ($R^2$ = 0.99, D' = 0.97), and is potentially a marker of the PD+RBD subgroup[32], which is a group likely to develop PD with dementia[3]. The same variant also colocalized with *SNCA-AS1* expression in the most recent DLB GWAS[8], strengthening evidence that this mechanism could be associated with cognitive decline. This hypothesized role of *SNCA-AS1* as an important, neuronally specific regulator of alpha-synuclein protein expression as a determinant of risk of alpha-synucleinopathies should be further studied.

We found a similar phenomenon in the *SCARB2* locus; rs7697073 is associated with RBD in the current GWAS, whereas in the recent PD GWAS there is an independent association at rs6825004. The PD-associated variant is possibly associated with *SCARB2* expression in the substantia nigra, while the RBD-associated variant, rs7697073, is not. This potential difference in *SCARB2* expression should be considered with caution, as the association with expression in the substantia nigra does not survive correction for multiple comparisons. We can hypothesize that if this difference will be proven to be true, it may lead to an earlier degeneration of the nigrostriatal fibers in the PD cohort compared to the RBD cohort, thus explaining the earlier manifestation of motor symptoms in the former. In RBD, the top-associated variant in this locus (rs7697073), like the *SNCA* locus variant, is potentially associated more with expression in cortical brain regions, providing additional support for our hypothesis, although here too this association does not survive Bonferroni correction. *SCARB2*, encoding the Scavenger Receptor Class B Member 2, is the transporter of glucocerebrosidase (encoded by *GBA*) from the endoplasmic reticulum to the lysosome[33]. It is possible that in PD this transport is affected by the variant associated with *SCARB2* expression in the substantia nigra. In

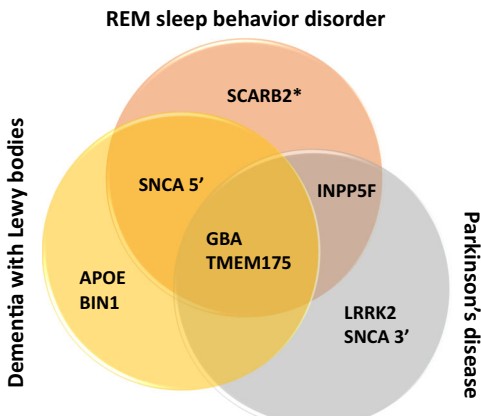

**Fig. 7 | Key GWAS-significant loci across three synucleinopathies.** It has been shown that the genetic risk for PD and DLB do not overlap completely, and we show that the same is true for RBD and the other two synucleinopathies. Here, we demonstrate key genetic risk loci for the three synucleinopathies. Only *GBA* and *TMEM175* are shared between all three, both of which play a role in the autophagy-lysosomal pathway. *SNCA* plays a role in PD, DLB, and RBD risk, yet the strongest risk locus for PD is at the 3' end of the gene while RBD and DLB share a risk locus at the 5' end. Similarly, *SCARB2* is a risk factor for PD as well as RBD, however, the RBD locus is independent of the variant identified for PD risk (as indicated by the asterisk in the figure).

RBD, this transport may be more affected in cortical regions. This could lead to specific vulnerabilities of specific brain regions in PD and RBD. For example, PD patients with RBD show significant cortical thinning when compared both to controls and to PD without RBD[34].

Despite the inclusion of PD patients with probable RBD in the current meta-analysis, notable PD and DLB GWAS loci are absent, including *LRRK2, MAPT, BIN1*, and *APOE*. Our sample size is comparable to PD[35] and DLB[8] GWAS that detected these signals (cases = 1713 & 2981, controls = 3978 & 4391, respectively). Examining the most significant variants in these loci, our study is sufficiently sized to detect these signals at GWAS significance with >80% power if they had comparable effects to their PD and DLB results. While this does not completely rule out association, it is evidence that these loci do not play as strong a role in iRBD or RBD-related PD. The apparent lack of association with RBD in these important regions, which we have previously reported in candidate gene studies in smaller cohorts[36–39], further supports RBD as a distinct subtype, genetically and clinically. These findings suggest that PD and DLB likely include different subgroups, some of which are associated with variants in *LRRK2, MAPT*, and *APOE*, while the subgroup defined by having iRBD prior to the onset of PD or DLB is not (Fig. 7). The presence of subgroups within PD is also evident in the PRS analyses, showing that RBD PRS better distinguished RBD cohorts compared to controls and has minimal predictive capability for PD–pRBD.

When considering the five loci associated with RBD in the current study, the autophagy-lysosomal pathway (ALP) seems to have a major role in RBD, like PD and DLB. However, uniquely to RBD, four of the five nominated genes in this GWAS directly engage with glucocerebrosidase (GCase) function. On top of the *GBA* locus itself, which encodes for GCase, alpha-synuclein has been shown to directly interact with GCase products and inhibit its transport to the lysosome[40,41]. Variants in the *SNCA* locus also seem to be modifiers of *GBA* penetrance and the age at onset (AAO) among *GBA* variant carriers[42]. *SCARB2* is the transporter that carries GCase to the lysosome, and *TMEM175* (encoding a lysosomal potassium channel) variants affect the activity of GCase in humans and in vitro[10]. Furthermore, *TMEM175* variants may affect the AAO of *GBA*-associated PD[26,42]. Although *INPP5F* does not have a known prominent role in the ALP, it may play a role in endosomal trafficking[43], and its neighboring gene *BAG3* is a moderator of selective autophagy for misfolded proteins and directly interacts with GCase[44,45]. At our sample size, we are not powered to detect the PD *BAG3* association or determine the true causal signal at this locus. Additionally, rare loss-of-function variants in *PSAP*, which encodes a co-activator of GCase (saposin C), are associated with risk for iRBD[46]. Rare mutations in the *PSAP* saposin D domain may also be associated with autosomal dominant PD[47], and a role for common *PSAP* variants in PD susceptibility is debated[47,48]. Taken together, these previous and current findings strongly highlight the role of GCase activity in RBD. Early PD GWAS with comparable sample sizes and DLB GWAS loci do not show the same specificity to the ALP or GCase function, and instead span multiple potential pathways and mechanisms. Since iRBD may appear years before the onset of overt neurodegeneration, this population is especially attractive for clinical trials aimed to prevent neurodegeneration in RBD-associated alpha-synucleinopathies (30–60% of PD[3], 50–80% of DLB[6]). *GBA* targeting therapies could also be tested in this population, especially those who carry *GBA* variants. Additionally, all loci nominated in this study are also linked to PD AAO via GWAS[26], which makes sense since both RBD and *GBA* are associated with more severe PD[49,50], the latter also associated with early-onset PD[51]. In this study, both iRBD and PD+pRBD populations are not below average in AAO (65 ± 8 and 69 ± 9, respectively), so we do not attribute our findings to simply an AAO association.

There are several limitations to this study. First, despite being the largest RBD cohort analyzed to date, this is still a relatively small GWAS, and future RBD GWASs will likely yield additional associations.

Power must be considered when interpreting these results, particularly in the lack of association of RBD PRS with PD without RBD, and in the genetic correlation analyses where we observe high variability for some traits. This issue may confound the LD-score regression results; however, we find it important to report these results as foundational, preliminary work in RBD genetics. Second, the meta-analysis may be PD-skewed since over half of the GWAS case cohort consists of PD+pRBD patients, some of which had iRBD prior to PD and some had PD symptoms first and RBD developed later. Given the different ascertainment method of the cases, there is evidence of heterogeneity between the two cohorts (notably at the *SNCA* signal, where the effect size is larger in iRBD than PD+pRBD, Table 1). It is therefore possible that some signals identified by the meta-analysis are PD-enriched, and that iRBD-specific signals are diluted. Yet, of all the variants analyzed in this study, less than 10% show heterogeneity $I^2 > 0.5$ with a $p$-value < 0.05. Additionally, the PD+pRBD cohort notably behaves differently (e.g., the *LRRK2* and *MAPT* loci) compared to the published PD GWAS[22], which includes both PD with and without RBD, suggesting the subset of PD patients who have RBD may be genetically distinct. The specific associations in *SNCA* and *SCARB2*, enrichment of the RBD GWAS loci in the ALP (unique compared to similarly powered PD or DLB GWAS), and lack of association between the RBD PRS and PD–RBD risk support that even with a PD-skewed population we are tagging distinct RBD risk variants. These points highlight the validity and importance of this study, but future studies of larger iRBD cohorts will be important to expand on these findings. A meta-analysis of only iRBD patients would be a better model, but since the iRBD cohort included in this study is the only cohort with genetic data available worldwide, increasing power with a pRBD cohort allowed interesting discoveries to be made. On this topic, we prioritized power for discovery[52] over an independent replication set, however each of the nominated loci have been linked to either PD or DLB, so it is highly unlikely these are random associations. Additionally, we find the same direction of effect for each nominated GWAS loci in the McGill PD+pRBD cohort, however it is underpowered to significantly detect these associations (Supplementary Table 1). Third, due to data protection in 23andMe, we could not examine for overlap of samples between the iRBD cohort and the PD+pRBD cohort. However, since less than 15% of the iRBD cohort has converted to PD, and since it is highly unlikely that all of them participated in 23andMe, if there is any overlap it is most likely minimal. Finally, this study only includes participants of European ancestry. Future studies in other populations are vital to characterize RBD genetic risk across all ancestries.

The results of the current study suggest that the genetic background of RBD, PD and DLB only partially overlap, and larger RBD studies will be required to better elucidate the genetic background of RBD. The present study also suggests that the lysosomal pathway, and more specifically the *GBA* pathway, could be a crucial target for therapeutic development targeting RBD and aimed to prevent neurodegeneration in this population.

## Methods
### Population
We used two cohorts for the RBD GWAS meta-analysis. The first is an iRBD cohort (N cases = 1061, N controls = 8386). The second is a cohort of PD patients with probable RBD (pRBD, N cases = 1782, age 69.1 ± 9 and N controls = 131,250, age 68.9 ± 9), genotyped and analyzed by 23andMe, Inc. pRBD was identified using the RBD Single-Question Screen (RBD1Q)[53], which has a sensitivity of 93.8% and specificity of 87.2% in PD. We did not select individuals with pRBD but without PD from 23andMe, since a small percentage of them will actually have iRBD; the questionnaire is not reliable for the general population but much more reliable for PD patients, albeit with false positives and false negatives. The meta-analysis combines the two for a total of 2843 cases and 139,636 controls. iRBD cases were collected by the International

RBD Study Group and were genotyped and analyzed at McGill University. This iRBD cohort included large cohorts of French, French Canadian, Italian and British origins, and smaller cohorts from different European populations. Accordingly, for controls, we used genotype data obtained from (a) French and French-Canadian controls from McGill University ($N = 871$); (b) the HYPERGENES Project[54] ($N = 557$ Italian samples); (c) the Wellcome Trust Case Control Consortium[55] ($N = 5516$ British samples); (d) European control samples genotyped in the Laboratory of Neurogenetics (LNG), National Institute on Aging (NIA), National institutes of Health (NIH) ($N = 1442$). Principal components to adjust for population substructure were used as mentioned below. The cases were aged $68 +/− 9$ years (standard deviation) on average and were 81% male, and the controls were aged $58.5 +/− 9$ years on average, 68% male. 23andMe cases and controls were age- and sex-matched, with 83% over 60 years of age and 64.5% male. In polygenic risk score (PRS) analyses, we used an independent replication cohort of PD +/−p RBD from McGill ($N$ cases = 502, $N$ controls = 907, average age $61 +/− 8$ years and 62% male) and 604 PD samples with recorded age at onset (AAO), including the 502 samples with pRBD data (average AAO $58 +/− 10$ years, 63% male). All cohorts used were confirmed European ancestry with principal component analysis using HapMap 3 as the reference population.

iRBD, referring to those who were diagnosed with RBD before developing overt neurodegeneration, was diagnosed according to the International Classification of Sleep Disorders (2nd or 3rd Edition), including video polysomnography. In the PD cohorts (except the 23andMe cohort, see below), PD was diagnosed by movement disorder specialists in accordance with the UK Brain Bank Criteria[56] or International Parkinson Disease and Movement Disorders Society[57] criteria. Within the PD cohorts, pRBD was identified using either the RBD single-question screen (RBD1Q)[53] or the RBD screening questionnaire (RBDSQ)[58], both with high sensitivity and specificity in PD[59]. All study participants signed informed consent forms, and the study protocol was approved by the institutional review boards.

## Genome-wide association study

**23andMe.** 23andMe cohorts were collected, genotyped, and filtered as previously described[9]. Briefly, all individuals included in the analyses provided informed consent and answered surveys online according to the 23andMe human subject protocol, which was reviewed and approved by Ethical & Independent Review Services, a private institutional review board (http://www.eandireview.com). DNA extraction and genotyping were performed on saliva samples by National Genetics Institute (NGI), a CLIA licensed clinical laboratory and a subsidiary of Laboratory Corporation of America. Samples were genotyped on one of five genotyping platforms. The v1 and v2 platforms were variants of the Illumina HumanHap550+ BeadChip, including about 25,000 custom SNPs selected by 23andMe, with a total of about 560,000 SNPs. The v3 platform was based on the Illumina OmniExpress+ BeadChip, with a total of about 950,000 SNPs. The v4 platform was a fully customized array, including about 570,000 SNPs. The v5 platform, in current use, is an Illumina Infinium Global Screening Array (~640,000 SNPs) supplemented with ~50,000 SNPs of custom content. Samples that failed to reach 98.5% call rate were re-analyzed. Individuals whose analyses failed repeatedly were re-contacted by 23andMe customer service to provide additional samples.

Participants were restricted to those of European ancestry determined through an analysis of local ancestry[60]. Briefly, a support vector machine (SVM) is used to classify individual haplotypes into one of 31 reference populations (https://www.23andme.com/ancestry-composition-guide/). The SVM classifications are then fed into a hidden Markov model (HMM) that accounts for switch errors and incorrect assignments, and gives probabilities for each reference population in each window. Finally, simulated admixed individuals are used to recalibrate

the HMM probabilities so that the reported assignments are consistent with the simulated admixture proportions.

GWAS was computed using logistic regression, assuming additive allele effects, adjusted for age, sex, genotype platform, and top 5 principal components (PCs). Imputed dosages were considered for imputed SNPs rather than best-guess genotypes. $p$-values were computed using a likelihood ratio test and adjusted for LD-score regression intercept to account for sample size mismatch between cases and controls.

**All other cohorts.** The iRBD cases and controls were genotyped on the OmniExpress GWAS chips (Illumina Inc.). We performed quality control according to a standardized GWAS pipeline (https://github.com/neurogenetics/GWAS-pipeline). At the sample level, individuals were screened for high or low heterozygosity ($−0.15 <= F <= 0.15$ for inclusion), call rate (>95%), genetic sex matching reported data ($0.25 < F < 0.75$) using plink 1.9. Ancestry outliers were detected using HapMap3 PCA data in R version 4.0.1. Samples identified with Asian, African, or mixed-race ancestry ($N = 139$) were removed. We used gcta64 to check for relatedness with other samples closer than cousin (pihat > 0.125). SNPs were filtered based on variant missingness (<0.05), disparate missingness between cases and controls ($p > 1E-04$), missingness by haplotype ($p > 1E-04$), and deviation from Hardy-Weinberg equilibrium in controls ($p > 1E-04$) using plink 1.9. To merge HYPERGENES, Wellcome Trust, and LNG controls with the McGill iRBD genotypes, we performed quality control on each cohort separately, then merged the cohorts using only variants common in all datasets. Following quality control, genotypes were filtered for minor allele frequency (MAF) > 0.01 to reduce imputation errors and imputed using Michigan Imputation Server and the Haplotype Reference Consortium[61] r1.1 2016 reference panel (GRCh37/hg19). Only imputed genotypes with an $R^2 > 0.30$ were kept for analysis, and imputed rare variants (MAF < 0.01) were excluded. Variants with MAF < 0.01 but directly genotyped and previously found to be associated with synucleinopathies (*GBA* variant p.N370S[12,42] and *LRRK2* variant p.G2019S[13], Supplementary Table 2) were included in the study.

GWAS was performed using rvtests[62] logistic regression with single Wald association test, including sex, age, and three ancestry PCs determined by scree plot for each cohort as covariates. We implemented METAL[63] to perform fixed-effect meta-analysis and FUMA[64] to identify top hits according to the standard GWAS $p$-value threshold of $p < 5E-08$. FUMA implemented an R2 threshold of 0.6 to define independent SNPs among those with GWAS-level significance, with a subsequent R2 threshold of 0.1 to define lead SNPs within LD blocks. To determine whether secondary associations were present in the different loci, we used GCTA-COJO[65] cojo-slct, a stepwise association method to identify independent associations, with default parameters. Power calculations for the meta-analysis are depicted in Supplementary Fig. 11.

## Polygenic risk score

PRSice2[66] and PLINK[67] 1.9 were used to calculate polygenic risk scores (PRS). To minimize overfitting, a portion of iRBD cases ($N = 212$) and controls ($N = 1265$) were withheld as a testing set. The GWAS and meta-analysis were redone excluding these samples. We set the p-value ceiling at the GWAS FDR significance threshold ($p < 1E-05$). Independent variants according to the standard pruning parameters defined by PRSice2 ($R^2 > 0.1$) passing this threshold ($N = 47$) were used to calculate the PRS. Using PRSice2, we calculated PRS for each individual as the average of effect sizes for each effect allele from the $N = 47$ risk profile found in the sample. We implemented receiver operating characteristic and area under the curve (AUC) analysis, using R version 4.0.1, to

determine the accuracy of this PRS in differentiating between cases and controls in iRBD (the withheld samples), an independent PD+pRBD cohort ($N = 285$), and a PD−pRBD cohort ($N = 217$) and controls ($N = 907$). Additionally, we used unadjusted linear regression to test whether polygenic risk for RBD is associated with RBD age at onset (AAO) or rate of conversion (AAO or rate of conversion ~ PRS), and Kaplan−Meier survival analysis to test whether groups divided into quartiles based on PRS convert significantly faster or slower than others. PRS code can be found on https://github.com/lynnekrohn/RBD_GWAS/blob/main/1_PRS.md.

## Pathway analysis

Pathway analysis was performed using functional enrichment analysis, specifically gene-set enrichment, using the publicly available online tool WebGestalt (http://www.webgestalt.org/)[68]. The gene-set enrichment analysis examines the enrichment of a provided set of genes in predetermined lists of genes involved in various functional pathways, detailed in Supplementary Data 3. An enrichment score is calculated, representing the level to which these genes are over-expressed in the various pathways, and then statistical significance is calculated using permutation testing. RBD genes included were those closest to the most significant GWAS SNP at the GWAS significance level, a single gene from each locus, as including multiple genes from the same locus may lead to false enrichment. We opted for choosing the nearest gene and not using QTLs, since in many loci there are multiple QTLs in multiple tissues with multiple genes, which will make the selection of genes for this preliminary analysis based on QTLs challenging. Multiple hypothesis adjustment is applied in accordance with the false-discovery rate (FDR).

## Colocalization

Coloc (version 4.0.1; https://github.com/chr1swallace/coloc)[69] was used to evaluate the probability of RBD loci and expression quantitative trait loci (eQTLs) sharing a single causal variant. Cis-eQTLs were derived from eQTLGen (accessed 19/02/2020; https://www.eqtlgen.org/cis-eqtls.html)[70] and PsychENCODE (accessed 20/02/2020; http://resource.psychencode.org/)[71], which represent the largest human blood and brain expression datasets, respectively (eQTLGen, $N = 31,684$ individuals; PyschENCODE, $N = 1387$ individuals). We additionally investigated eQTLs from individual CNS cell types (astrocytes, endothelial, microglia, oligodendrocytes and precursors, and pericytes) using eQTLs generated by Bryois et al. (https://malhotralab.shinyapps.io/brain_cell_type_eqtl/)[22]. For each locus, we examined all genes within 1 Mb of a significant region of interest, as defined by RBD ($p < 5 \times 10^{-8}$). Coloc was run using default $p_1 = 10^{-4}$ and $p_2 = 10^{-4}$ priors ($p_1$ and $p_2$ are the prior probability that any random SNP in the region is associated with trait 1 and 2, respectively). The $p_{12}$ prior (the prior probability that any random SNP in the region is associated with both traits) was altered to $p_{12} = 5 \times 10^{-6}$, which has been shown to be a more robust choice than the default $p_{12} = 10^{-5}$[72]. Loci with a posterior probability of hypothesis 4 (PPH4) $\geq 0.8$ were considered colocalized due to a single shared causal variant, as opposed to two distinct causal variants (PPH3). All colocalizations were subjected to sensitivity analyses using coloc's sensitivity() function, which plots prior and posterior probabilities of each coloc hypothesis as a function of the $p_{12}$ prior. This permits exploration of the robustness of our conclusions to changes in the $p_{12}$ prior. Code for coloc analyses is openly available at https://github.com/RHReynolds/RBD-GWAS-analysis/.

## Cell-type and tissue specificity measures

Specificity represents the proportion of a gene's total expression attributable to one cell type/tissue, with a value of 0 meaning a gene is not expressed in that cell type/tissue and a value of 1 meaning that a gene is only expressed in that cell type/tissue. To determine specificity of a gene to a tissue or cell-type, specificity values from two independent gene expression datasets were generated, as described in Chia et al.[8]. Briefly, these datasets included (1) bulk-tissue RNA-sequencing of 53 human tissues from the Genotype-Tissue Expression consortium (GTEx; version 8)[20] and (2) human single-nucleus RNA-sequencing of the middle temporal gyrus from the Allen Institute for Brain Science (AIBS; https://portal.brain-map.org/atlases-and-data/rnaseq/human-mtg-smart-seq)[21]. Specificity values for GTEx were generated using modified code from a previous publication (https://github.com/jbryois/scRNA_disease)[73], and modified to reduce redundancy among brain regions and to include protein- and non-protein-coding genes. Expression of tissues was averaged by organ (except in the case of brain). Thus, specificity values were generated for a total of 35 tissues. Specificity values for the AIBS-derived dataset were generated using gene-level exonic reads and the "generate.celltype.data" function of the EWCE package (https://github.com/NathanSkene/EWCE)[74]. Specificity values for both datasets and the code used to generate these values are openly available at: https://github.com/RHReynolds/MarkerGenes.

## Heritability & genetic correlation

Heritability of RBD and shared heritability across traits (genetic correlation) were calculated in clinically confirmed cases of iRBD using linkage-disequilibrium (LD) score regression on LDHub (http://ldsc.broadinstitute.org/ldhub/)[75,76]. Traits for shared heritability tests were chosen based on previous association to a synucleinopathy (e.g., smoking behaviors, education levels) and relevance to RBD (e.g., sleep disorders). Owing to the limited sample sizes in this GWAS, we chose a hypothesis-driven genetic correlation study rather than an unbiased approach using all available LDHub traits. Summary statistics for the compared traits were accessed through the LDHub platform or downloaded from publicly available sources, then formatted and analyzed using LDHub python v2.7 scripts. Bonferroni correction was calculated based on the number of traits tested ($N = 27$, $p < 0.0019$). LD-score regression code is available on https://github.com/lynnekrohn/RBD_GWAS/blob/main/4_LD-regression.md.

## IRB statement

Participants provided informed consent and participated in the research online, under a protocol approved by the external AAHRPP-accredited IRB, Ethical & Independent Review Services (E&I Review) by the REB of the Montréal Neurological Institute. Participants were included in the analysis on the basis of consent status as checked at the time data analyses were initiated.

## Reporting summary

Further information on research design is available in the Nature Portfolio Reporting Summary linked to this article.

# Data availability

The iRBD summary statistics are publicly available on GWAS catalog (https://www.ebi.ac.uk/gwas/, study accession GCST90204200). The full GWAS summary statistics for the 23andMe discovery dataset will be made available through 23andMe to qualified researchers under an agreement with 23andMe that protects the privacy of the 23andMe participants. Please visit https://research.23andme.com/collaborate/#dataset-access/ for more information and to apply to access the data. The GWAS summary statistics for traits analyzed for genetic correlation with RBD can be found on LDHub (http://ldsc.broadinstitute.org/ldhub/) or GWAS Catalog (https://www.ebi.ac.uk/gwas/). The quantitative trait loci data used for fine-mapping can be accessed on: eQTLGen https://www.eqtlgen.org/cis-eqtls.html; PsychENCODE http://resource.psychencode.org/; GTEx v8 https://www.gtexportal.org/home/; AIBS https://portal.brain-map.org/atlases-and-data/rnaseq/human-mtg-smart-seq; and Bryois et al. CNS-specific cell types https://malhotralab.shinyapps.io/brain_cell_type_eqtl/.

## Code availability

The code for all analyses are publicly available on https://github.com/lynnekrohn/RBD_GWAS/releases/tag/October2022[77] except for colocalization analyses, which can be found on https://github.com/RHReynolds/RBD-GWAS-analysis/, including adaptations of previously published code (https://github.com/jbryois/scRNA_disease and https://github.com/NathanSkene/EWCE).

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

## Acknowledgements

HYPERGENES Project: https://cordis.europa.eu/project/rcn/86758_en.html Wellcome Trust Case Control Consortium: https://www.wtccc.org.uk/. We thank the members of the NINDS Neurodegenerative Diseases Research Unit and the NIA Laboratory of Neurogenetic for their collegial support and technical assistance. We would like to thank the research participants and employees of 23andMe for making this work possible This work was financially supported the Michael J. Fox Foundation (MJFF-020700: Z.G-O.), Fonds de recherche du Québec (Doctoral Training Award: L.K.), and Parkinson Canada (Pilot Project Grant: L.K.). It was additionally supported by the Canadian Consortium on Neurodegeneration in Aging (CCNA), and the Canada First Research Excellence Fund (CFREF) awarded to McGill University for the Healthy Brains for Healthy Lives (HBHL) program, and was supported in part by the Intramural Research Program of the National Institute on Aging (NIA), National Institutes of Health, Department of Health and Human Services (project number ZO1 AG000535), as well as the National Institute of Neurological Disorders and Stroke.

## Author contributions

L.K., C.B., M.A.N., A.B. Singleton, and Z.G-O. conceptualized and supervised the study. J.A.R., F.A., and K.F. performed sample

preparations and validation. L.K. and K.H. performed data preparation, quality control, and genome-wide association studies, C.B. contributed to these analyses. L.K. performed meta-analysis and downstream analyses, K. Senkevich, U.R., M.A.E., E.Y., and K.B. offered data checks and support. R.H.R. performed colocalization, E.G. contributed and M.R. supervised. S.B.-C. and A.N. supervised Mendelian Randomization studies. I.A., M.T.M.H., J.Y.M., J.-F.G., A.D., Y.D., G.L.G., M. Viaene., F.J., A.B., B.H., A. Stefani., A.I., K. Sonka, D.K., W.O., A.J., G.P., F.B., E.A., M.F., M.P., B.M., B.H., C.T., F-S.D., V.C.C., C.C.M., A.H., L.F-S., F.D., M. Valente, B.A., and B.B. provided biospecimens and clinical data. P.C., R.C., S.W.S., G.A.R., L.P., R.B.P., M.A.N., and A. B. Singleton provided data, equipment, and supervisory support. L.K. generated figures and wrote the initial manuscript with Z.G-O. All authors critically reviewed and edited the article.

## Competing interests

S.W.S. serves on the Scientific Advisory Council of the Lewy Body Dementia Association. S.W.S. is an editorial board member for JAMA Neurology and the Journal of Parkinson's Disease. I.A. was previously consultant for Idorsia pharma, and UCB Pharma. A.D. served on the scientific advisory board for Eisai, UCB, Jazz Pharma, received research support from Jazz Pharma, Flamel Ireland, Canopy Growth, and honoraria from speaking engagements from Eisai and Sunovion. M.A.N.'s participation in this project was part of a competitive contract awarded to Data Tecnica International LLC by the National Institutes of Health to support open science research, he also currently serves on the scientific advisory board for Clover Therapeutics and is an advisor to Neuron23 Inc as a data science fellow. Z.G.O. is supported by the Fonds de recherche du QuebecSante (FRQS) Chercheurs-boursiers award, and is a Parkinson's Disease Canada New Investigator awardee. He received consultancy fees from Ono Therapeutics, Handl Therapeutics, Neuron23, Lysosomal Therapeutics Inc., Bial Biotech Inc., Deerfield, Lighthouse, and Idorsia, all unrelated to the current study. K.H., P.F., and the 23andMe Research Team are employed by and hold stock or stock options in 23andMe, Inc. The remaining authors declare no competing interests.

## Additional information

[1]Department of Human Genetics, McGill University, Montréal, QC, Canada. [2]The Neuro (Montreal Neurological Institute-Hospital), McGill University, Montréal, QC, Canada. [3]23andMe, Inc., Sunnyvale, CA, USA. [4]Laboratory of Neurogenetics, National Institute on Aging, National Institutes of Health, Bethesda, MD, USA. [5]Department of Neurodegenerative Disease, UCL Queen Square Institute of Neurology, University College London, London, UK. [6]Great Ormond Street Institute of Child Health, Genetics and Genomic Medicine, University College London, London, UK. [7]NIHR Great Ormond Street Hospital Biomedical Research Centre, University College London, London, UK. [8]Lund University, Translational Neurogenetics Unit, Department of Experimental Medical Science, Lund, Sweden. [9]Sleep Disorders Unit, Pitié Salpêtrière Hospital, APHP-Sorbonne, Paris Brain Insitute and Sorbonne University, Paris, France. [10]Oxford Parkinson's Disease Centre (OPDC), University of Oxford, Oxford, UK. [11]Nuffield Department of Clinical Neurosciences, University of Oxford, Oxford, UK. [12]Centre d'Études Avancées en Médecine du Sommeil, Hôpital du Sacré-Cœur de Montréal, Montréal, QC, Canada. [13]Department of Psychiatry, Université de Montréal, Montréal, QC, Canada. [14]Department of Psychology, Université du Québec à Montréal, Montreal, QC, Canada. [15]Department of Neurosciences, Université de Montréal, Montréal, QC, Canada. [16]National Reference Center for Narcolepsy, Sleep Unit, Department of Neurology, Gui-de-Chauliac Hospital, CHU Montpellier, University of Montpellier, Institute Neuroscience Montpellier Inserm, Montpellier, France. [17]Clinical Neurology Unit, Department of Neurosciences, University Hospital of Udine, Udine, Italy. [18]Department of Medicine (DAME), University of Udine, Udine, Italy. [19]Sleep Disorders Clinic, Department of Neurology, Medical University of Innsbruck, Innsbruck, Austria. [20]Department of Neurology and Centre of Clinical Neuroscience, Charles University, First Faculty of Medicine and General University Hospital, Prague, Czech Republic. [21]Department of Neurology, Philipps-University, Marburg, Germany. [22]Department of Biomedical, Metabolic and Neural Sciences, University of Modena and Reggio-Emilia, Modena, Italy. [23]IRCCS, Institute of Neurological Sciences of Bologna, Bologna, Italy. [24]Department of Biomedical and Neuromotor Sciences (DIBINEM), Alma Mater Studiorum, University of Bologna, Bologna, Italy. [25]Department of Neurosciences, Biomedicine and Movement Sciences, University of Verona, Verona, Italy. [26]Department of Medical Sciences and Public Health, Sleep Disorder Research Center, University of Cagliari, Cagliari, Italy. [27]Paracelsus-Elena-Klinik, Kassel, Germany. [28]Department of Neurology, University Medical Centre Goettingen, Goettingen, Germany. [29]Sleep and Neurology Unit, Beau Soleil Clinic, Montpellier, France. [30]EuroMov Digital Health in Motion, University of Montpellier IMT Mines Ales, Montpellier, France. [31]University Lille North of France, Department of Clinical Neurophysiology and Sleep Center, CHU Lille, Lille, France. [32]Institute of Sleep Medicine and Neuromuscular Disorders, University of Münster, Münster, Germany. [33]Department of Neurological Sciences, Università Vita-Salute San Raffaele, Milan, Italy. [34]Laboratory for Sleep Disorders, St. Dimpna Regional Hospital, Geel, Belgium. [35]Department of Neurology, St. Dimpna Regional Hospital, Geel, Belgium. [36]Department of Neurology, Antwerp University Hospital, Edegem, Belgium. [37]Sleep disorder Unit, Carémeau Hospital, University Hospital of Nîmes, Nîmes, France. [38]Department of Neurology, Mayo Clinic, Rochester, MN, USA. [39]Neurodegenerative Diseases Research Unit, National Institute of Neurological Disorders and Stroke, Bethesda, MD, USA. [40]Department of Neurology, Johns

Hopkins University Medical Center, Baltimore, MD, USA. [41]Preventive Neurology Unit, Wolfson Institute of Preventive Medicine, Queen Mary University of London, London, UK. [42]Department of Clinical and Movement Neurosciences, University College London, Institute of Neurology, London, UK. [43]Department of Neurology, Oslo University Hospital, Oslo, Norway. [44]Data Tecnica International, Glen Echo, MD, USA. [45]Center for Alzheimer's and Related Dementias, National Institutes of Health, Bethesda, MD, USA. [46]Department of Neurology and Neurosurgery, McGill University, Montreal, QC, Canada. *A list of authors and their affiliations appears at the end of the paper. ✉e-mail: ziv.gan-or@mcgill.ca

## 23andMe Research Team

Stella Aslibekyan[3], Adam Auton[3], Elizabeth Babalola[3], Robert K. Bell[3], Jessica Bielenberg[3], Katarzyna Bryc[3], Emily Bullis[3], Daniella Coker[3], Gabriel Cuellar Partida[3], Devika Dhamija[3], Sayantan Das[3], Sarah L. Elson[3], Teresa Filshtein[3], Kipper Fletez-Brant[3], Pierre Fontanillas[3], Will Freyman[3], Pooja M. Gandhi[3], Barry Hicks[3], David A. Hinds[3], Ethan M. Jewett[3], Yunxuan Jiang[3], Katelyn Kukar[3], Keng-Han Lin[3], Maya Lowe[3], Jey C. McCreight[3], Matthew H. McIntyre[3], Steven J. Micheletti[3], Meghan E. Moreno[3], Joanna L. Mountain[3], Priyanka Nandakumar[3], Elizabeth S. Noblin[3], Jared O'Connell[3], Aaron A. Petrakovitz[3], G. David Poznik[3], Morgan Schumacher[3], Anjali J. Shastri[3], Janie F. Shelton[3], Jingchunzi Shi[3], Suyash Shringarpure[3], Vinh Tran[3], Joyce Y. Tung[3], Xin Wang[3], Wei Wang[3], Catherine H. Weldon[3], Peter Wilton[3], Alejandro Hernandez[3], Corinna Wong[3] & Christophe Toukam Tchakouté[3]

