## [Peer Review File · Nature Communications]

Genome-wide association study of REM sleep behavior disorder identifies polygenic risk and brain expression effectsREVIEWER COMMENTS

Reviewer #1 (Remarks to the Author):

This is a very comprehensive study of the genetic component of REM sleep behavior disorder (RBD). In addition to the first genome-wide association study (GWAS), expression, colocalization, PRS and pathway analysis provide a complete overview of the genetic component of RBD. Five loci were identified associated to RBD, with 2 of them (SCARB2 and INPP5F) being new. Interestingly all 5 of them have been also implicated in PD, although some of the SNPs causing the association (in SNCA and SCARB2) are different and not in LD with the top-associated SNPs in PD. The authors suggest, based on these differences, together with differences in colocalization, PRS and LD-score regression analysis, that the genetic background of RBD only partially overlaps with PD and DLB.

This is an extremely well written article in a very much needed topic. My main concern with this manuscript is that I am not completely convinced that RBD is a distinct entity as there is substantial overlap. In fact all 5 genes, as the authors point out, have been previously associated with PD. Based on their results RBD resembles GBA-associated PD, and in fact one of the GBA mutations is their top hit in this gene. I believe the authors should be more cautious on their conclusion, and explore other possible hypothesis on how RBD and PD could be linked.

Some additional minor comments:

– I know there is no much room in the manuscript, but it would be important for the readers to know what do the 20% individuals with RBD who do not develop neurological disorders look like. For example, do they also have synuclein aggregates? Perhaps just a mention and cite a few articles if available.

– Just a formatting issue: Something is off with the period like in line 155, 12.3%, and the following standard error 0.07, the period is higher than it should be

– I wonder if the negative result with PRS and PD-RBD could be due to a lack of statistical power instead of tagging RBD-specific loci.

– Studies in SNCA show there are at least 2, maybe even 3 independent signals coming from the gene and risk of PD. Authors should mention if rs3756059 is in LD with any of the 5' SNPs associated to PD.

– As I mentioned at the top, one of my main issue with the manuscript is the lack of proof towards RBD being a distinct genetic entity. As an example, the statement in the discussion: "A distinct genetic background for RBD is also supported by the lack of genetic correlation between iRBD and the most recent PD GWAS..." is not justified as there is quite a lot of overlap between both as shown in Figure 5A, and it is only in a few loci that the direction is different between the two.

– The authors should mention if the check for any association with BST1, or LAMP3 in their GWAS as they recently published an association between RBD and variants in these 2 genes. I am surprised they did not even cite their work.

– The authors should mention what the LD is between the two GBA variants in Table 1.

Reviewer #2 (Remarks to the Author):

My comments to authors are in the attached PDF document.

GWAS of RBD identifies novel loci with distinct polygenic and brain expression effects

The authors report the results of the first GWAS of RBD, with the identification of 6 independent associations in 5 genome-wide significant susceptibility loci, each labeled with the symbol of the nearest gene (SNCA, GBA, TMEM175, INPP5F and SCARB2). The authors also conducted post-GWAS (coloc, geneset enrichment, polygenic risk score, and genetic correlation) analyses and, based on their results, claim RBD is a genetically distinct alpha-synucleinopathy subtype.

RBD is described as either 1) a clinical symptom that precedes diagnosis of alpha-synucleinopathies [mainly PD and DLB] by 10+ years [referred to as iRBD], or 2) a clinical symptom that rather frequently manifests itself after diagnosis of alpha-synucleinopathies [referred to as sRBD, or probable RBD (pRBD) when ascertained via a questionnaire].

The authors specifically describe RBD as a “more malignant subtype of alpha-synucleinopathies” and iRBD as a “prodromal alpha-synucleinopathy”.

Specifically, the authors performed two GWASes, followed by meta-analysis:

- 1) A GWAS cohort [iRBD] with 1,061 iRBD cases and 8,386 controls (no iRBD)
- 2) A GWAS cohort [23andMe] with 1,782 PD+pRBD cases and 131,250 controls (no PD and no iRBD)

RBD ascertainment differs for the two cohorts (clinical diagnosis vs. self-reported based on single question).

Here are my comments related to each subsection of the Results section, followed by some minor issues and a general one, which - in my opinion - would need to be addressed prior to publication in Nature Communications:

GWAS

- For the ease of the reader, in Methods - GWAS, please add a brief description of how the 23andMe cohort was collected, genotyped and filtered in addition to citing previous work. More importantly, it is not clear how the 23andMe GWAS was performed. Were variants imputed and rvttests used as described for “All other cohorts”? If done differently please provide a detailed description of the procedures used. If NOT done differently, please make it clear in the text.
- It is mentioned that rare genotyped variants were included if previously shown to be associated with alpha-synucleinopathies. The full list of these variants and their QC parameters, in particular MAC in cases and controls, and differential missingness test statistics for each GWAS cohort, should be included in the supplementary materials.
- In addition to the above, the authors should also explain how error rates due to the large imbalance in cases and controls in both cohorts (but particularly severe for the 23andMe cohort) were controlled.

- One point highlighted by the authors in the title/abstract and main text is that RBD is genetically distinct from PD and DLB based, in part, on the lack of significant associations at well-established PD and DLB GWAS loci in their RBD GWAS. However, as mentioned by the authors themselves, lack of statistical significance does not prove the null hypothesis and, in an admittedly small GWAS study as the one described by the authors, lack of significant association would indeed be a common occurrence even in the presence of a true effect. Indeed, this has been observed throughout the history of GWAS for several traits and diseases as sample sizes increased over time. In addition, since statistical power is mentioned by the authors repeatedly, it would be useful to include the results of power analyses of their RBD GWAS (iRBD, PD+pRBD, and meta) for MAFs and effect sizes commonly observed at PD and DLB GWAS loci.
- In support of their hypothesis that RBD is genetically distinct from PD and DLB, the authors also point to RBD association signals at two loci (SNCA and SCARB2) being independent from PD association signals at the same loci due to lack of LD between top SNPs. To further support this point, the results of conditional analyses at these two loci should be included in supplementary materials. I would also like to see a conditional analysis/comparison of RBD GWAS association signals (iRBD, PD+pRBD, and meta) with those in PD AAO GWAS loci discovered by the same authors, since RBD is mentioned to be associated with a more malignant subtype of alpha-synucleinopathies. Indeed, all 5 RBD loci are also PD AAO loci and the distinct genetic architecture of RBD susceptibility could be related to the one observed by the same authors between PD susceptibility and AAO. A conditional analysis/comparison with recently reported DLB GWAS loci should also be included, since iRBD can progress to both PD and DLB.
- The authors should also perform a GWAS comparing PD cases with and without pRBD in the 23andMe cohort, in order to identify loci specific to RBD susceptibility in PD. The authors should also consider performing a GWAS based on multinomial logistic regression to include all classes (PD+pRBD, PD-pRBD, and controls, and possibly also iRBD) and thus increase power and more rigorously identify loci with shared or distinct effects across different disease subtypes (see for example <https://doi.org/10.1093/bioinformatics/btw075> and <https://doi.org/10.1002/gepi.20486>).
- One major limitation of this GWAS is lack of replication in an independent cohort. However, an independent cohort including PD+pRBD (N=285), PD-pRBD (N=217) and controls (N=900 or 907 is reported at two different places in the text, this discrepancy should be resolved) was used for PRS studies. Therefore, replication (and subsequent meta-analysis) of genome-wide and suggestive associations identified by the PD+pRBD vs controls GWAS (performed by the authors) and the PD+pRBD vs PD-pRBD GWAS (suggested above) is possible and should be performed.

PRS

- This section should be revised in light of results from the PD+pRBD vs PD-pRBD or multinomial GWAS mentioned earlier. The ability of the RBD PRS to differentiate between PD+pRBD and PD-pRBD cases should also be assessed. Indeed, the title of this section points to the distinction between iRBD (or PD+pRBD) and PD-pRBD, yet (if I

understood correctly, given that the predicted outcomes are clearly indicated only for the iRBD AUC) the AUCs reported in the main text are between iRBD (or PD+pRDB or PD-pRDB) and controls.

- Is the RBD PRS associated with PD AAO?

DGE

- This section focuses on the comparison between the sizes, direction and p-values of eQTL effects of RBD and PD-associated variants in the SNCA and SCARB2 loci across several brain regions. The stated purpose of this analysis is to test the hypothesis that the different disease-associated SNPs have heterogeneous effects on regional gene expression in the brain. First, I find the titles of this section and of Fig. 2 a bit misleading, as they seem to refer to differential gene expression (DGE) rather than differential eQTL effects across brain regions. Indeed what's shown in the figure and presented in the text are eQTL effects of SNPs, not differential gene expression of SNCA and SCARB2 in different brain regions. Please clarify this in both titles and change "GTEX v8 Expression" in Fig. 2 to "GTEX v8 eQTL effects". Second, the only pairwise eQTL effects that survive multiple testing correction are observed for SNCA-AS1 (this should be noted in the text, in particular in the Discussion when claiming rs7697073 is associated with SCARB2 expression). While it is true that more statistically significant SNCA-AS1 eQTL associations are observed for the RBD variant compared to the PD variant, from this visual comparison it is impossible to assess the statistical significance (or simply the actual significance) of this difference, given that the direction of effect is consistent across the RBD and PD variants and no permutation (or otherwise valid NHST) p-values are provided for the actual hypothesis. This should be resolved, since the results of this analysis are one the main conclusions highlighted in title of the manuscript.
- The authors mentioned using multiple testing correction as done by GTEX; however, this should be better described in the manuscript since multiple testing correction depends on the hypothesis being tested and GTEX procedure may not be relevant here.
- If the authors firmly believe that this visual presentation is critical to support the main conclusion of the manuscript, then the same comparisons should be made between RBD [and iRBD] and DLB-associated variants, and ideally also between RBD [and PD+pRDB] and PD:AAO-associated variants. This would substantiate the authors claim that RBD is distinct from PD and DLB, and begin to investigate the possibility that RBD vs PD distinction may (or may not) be largely due to a faster progression rate in PD+pRDB vs PD-pRDB.
- I would also consider rewording the rather confusing title of this section because it is unclear to me how the analyses presented in this section relates to the fact that "differential gene expression in different brain regions may drive the independent associations of SNCA and SCARB2 [locus SNPs] with RBD and PD". If by "driving" the authors mean "mediating" then conditional SMR and/or PrediXcan-like analyses should be performed instead.

Coloc

- Also in this set of analyses, evidence of colocalization is observed only at the SNCA locus but not at the SCARB2 locus. This should be noted in the text, in particular in the Discussion when claiming rs7697073 may be acting on disease risk via modulation of SCARB2 expression.
- Given the interesting association between SNCA locus SNPs and MMRN1 expression in the blood and the specificity of MMRN1 expression in microglia of the brain, it would be interesting to perform colocalization analyses using myeloid and/or microglia-specific eQTL datasets that are publicly available (see for example <https://doi.org/10.1038/s41588-021-00976-y>).

Pathway analysis

- It is unclear how the list of candidate causal genes for pathway analysis was generated and what genes were included in it. In the Methods section the authors state: “RBD genes included were those closest to the most significant GWAS SNP” which is not a very precise description of the selection process that was used. Does this mean that genes within a certain distance range from a single SNP (the most significant GWAS SNP) were selected? Or (most likely) that, for each locus, only one gene (the one closest to the lead SNP in each locus) was selected? Were only genome-wide significant loci (hence only 5 genes) considered or also loci with suggestive evidence of association? More details are needed to better understand what was actually done.
- The authors should also justify why the nearest-gene criterion was utilized rather than, for example, nominating genes based on eQTL effects, or including all genes within or near the association region, for example. The example of BAG3 highlighted by the authors should, by itself, justify trying a different approach in addition to nearest-gene criterion.
- The supplementary table should also include the ID/Symbols of the genes that belong to each geneset or at least which of the GWAS-nominated genes are part of each geneset, in addition to the geneset size.

Comparison of PD with DRB GWAS loci

- Please add a panel to Fig. 5 to show the “PD vs RBD” (i.e., meta of iRBD and PD+pRBD) comparison.
- Are all the DLB GWAS-significant variants also included in this list of PD GWAS loci? Please generate additional figure and supplementary table like the ones shown, but using DLB instead of PD GWAS sumstats.

Heritability and genetic correlation

- The lack of sufficient power to accurately measure genetic correlation between iRBD and DLB is mentioned. Please describe how power was calculated and what its value was.

- Please discuss how this issue of limited power of iRBD GWAS to measure genetic correlations with other traits may confound the interpretation of the LD Score results as providing evidence for a distinct genetic architecture of iRBD.
- Please correct the claim that iRBD is genetically correlated with type II diabetes, since the correlation is not significant after multiple testing correction.
- In light of the above it is surprising that the positive (and almost significant after multiple testing correction) correlation between RBD and Alzheimer's disease was not discussed.
- Given that, in previous sections, the authors claimed distinct genetic architecture of RBD compared to PD based on distinct association signals at a couple of loci, and yet the results of these global genetic correlation studies are inconclusive and at the same time unsurprising, the authors' claim would be strengthened by performing local/locus-specific genetic correlation analyses between RBD (iRBD, PD+pRBD, and meta) and PD/DLB at those loci.

Minor issues

- In Methods - Population, please report average age + standard deviation of 23andMe cohort, separately for cases and controls.
- Describe how individuals in both cohorts were confirmed of European ancestry and ancestry outliers/mixed ancestry individuals removed using HapMap 3 as reference (visual inspection of PC plot? distance from ref pop centroid? global ancestry estimation?). If visual inspection was used, please include PC plot(s) in supplementary materials.
- QQ plot of individual RBD GWASes and meta-analysis should be included in supplementary materials.
- The locus labels are very difficult to read in Fig. 1A, please consider changing the color scheme.
- For the ease of the reader, please include a table where the sumstats across all disease traits of all RBD, iRBD, PD+pRBD, PD, PD:AAO and DLB-associated variants discussed in the main text are presented together.

General issue

While the authors do a good job of highlighting the limitations of their study (e.g., limited size) and briefly point to lack of statistical significance as unable to prove absence of an effect, often their claims in the discussion may still lead the reader down the path of the most common mis-interpretation of non-significant p-values in the biomedical literature, i.e., as evidence of a lack of effect. The authors should revise their writing in order to avoid this confusion and should also include effect size estimates and confidence intervals whenever reporting or citing an effect in their manuscript. Alternatively, the authors could use statistical methods other than NHST (e.g., Bayesian inference) to calculate $P(H_0 | \text{data})$ under a clearly stated set of assumptions.

We would like to thank the reviewers for the thorough review and helpful comments. Below we provide a response (in regular font) for each of the reviewers' comments (in bold). Thank you for considering our revised manuscript for publication in *Nature Communications*.

Reviewer #1

This is a very comprehensive study of the genetic component of REM sleep behavior disorder (RBD). In addition to the first genome-wide association study (GWAS), expression, colocalization, PRS and pathway analysis provide a complete overview of the genetic component of RBD. Five loci were identified associated to RBD, with 2 of them (SCARB2 and INPP5F) being new. Interestingly all 5 of them have been also implicated in PD, although some of the SNPs causing the association (in SNCA and SCARB2) are different and not in LD with the top-associated SNPs in PD. The authors suggest, based on these differences, together with differences in colocalization, PRS and LD-score regression analysis, that the genetic background of RBD only partially overlaps with PD and DLB. This is an extremely well written article in a very much needed topic. My main concern with this manuscript is that I am not completely convinced that RBD is a distinct entity as there is substantial overlap. In fact all 5 genes, as the authors point out, have been previously associated with PD. Based on their results RBD resembles GBA-associated PD, and in fact one of the GBA mutations is their top hit in this gene. I believe the authors should be more cautious on their conclusion, and explore other possible hypothesis on how RBD and PD could be linked.

We thank the reviewer for the opportunity to better explain this point. The word distinct is perhaps a bit strong (and we toned it down in the revised manuscript), because obviously many RBD patients convert to PD. The point we were making is that the genetic background of RBD represents a subtype that only partially overlaps with that of PD, and this is based on two observations: **1)** As the reviewer noted, there were two hits in PD loci (*SNCA* and *SCARB2*) where the associated variants were different, independent of those associated with PD, and importantly, have an effect in the opposite direction in RBD and PD, and **2)** There are known associations with PD and DLB that despite enough power, were not associated with RBD at all (*LRRK2*, *MAPT*, *APOE* and *BINI* – below we show the power to detect those at a GWAS significance level, which is >94% for all three loci). These two observations clearly demonstrate that also from the genetic point of view (on top of the clinical), RBD potentially represents an important subtype. As the reviewer noted, this is also supported by the downstream analyses we performed with LDSC and PRS.

We now better explain it in the first paragraph of the discussion, where it now reads (page 11, lines 349-366):

“In this first GWAS of RBD, we identified six RBD-associated loci in five genomic regions: two novel loci near SCARB2 and INPP5F and three previously reported loci near SNCA, GBA and TMEM175. Two of the loci, SNCA and SCARB2, have different and independent variants associated with RBD than those associated with PD. Our sample size had sufficient power (>80%) to detect variants

previously reported in PD and DLB at the GWAS significance level, namely, MAPT (rs62053943), LRRK2 (rs34637584), BIN1 (rs6733839) and APOE (rs769449)^{8,14}, yet there was no association. This does not provide a definite proof that these variants are not associated with RBD, yet it further suggests a genetic background in the RBD subtype that is only partially overlapping with those of PD and DLB as a whole. Colocalization analyses suggest that the SNCA variants are associated with differential expression of SNCA-AS1 in different brain regions, tissues, and cell types. We further show that RBD-specific PRS better predicts RBD case status than PD-pRBD when compared to controls. Pathway analysis and comparisons between this RBD meta-analysis and previous PD and DLB GWAS show specificity to the autophagy lysosomal pathway (ALP) in risk for iRBD and RBD-associated PD, whereas that specificity is not noted in similarly powered PD or DLB GWAS. Based on these results, we hypothesize that RBD is an indicator of a specific synucleinopathy subtype which is characterized by ALP dysfunction and cognitive decline, with clinical and genetic distinctions which should be considered in future drug development and clinical trials.”

Power analysis for the top PD/DLB-associated variants in *LRRK2*, *MAPT*, *APOE* and *BIN1* are presented below. Our study had enough power at GWAS significance to identify the associations reported in PD (for *LRRK2* and *MAPT*) and in DLB (*APOE* and *BIN1*), and there was clearly no association, again supporting RBD having a specific genetic background.

LRRK2 rs34637584 (p.G2019S)

Sample Size Cases/Controls = 0.028

Cases: 2843

Controls: 100000

Study Design

Significance Level: 0.00000050

Disease Model Additive

Prevalence: 0.0100

Disease Allele Frequency: 0.0015

Genotype Relative Risk: 10.0000

Results

Expected power for a one-stage study	0.998
Expected disease allele frequency	
Cases	0.0015
Controls	0.0015
Probability of disease	
Genotype A/A [with frequency 0.000]	0.185
Genotype A/B [with frequency 0.003]	0.097
Genotype B/B [with frequency 0.997]	0.018

MAPT rs62053943

Sample Size Cases/Controls = 0.028

Cases: 2843

Controls: 100000

Study Design

Significance Level: 0.00000050

Disease Model Additive

Prevalence: 0.0100

Disease Allele Frequency: 0.1552

Genotype Relative Risk: 1.3100

Results

Expected power for a one-stage study	0.948
Expected disease allele frequency	
Cases	0.192
Controls	0.155
Probability of disease	
Genotype A/A [with frequency 0.024]	0.000
Genotype A/B [with frequency 0.262]	0.000
Genotype B/B [with frequency 0.714]	0.000

APOE rs769449

Cases/Controls = 0.028

Sample Size

Cases: 2843

Controls: 100000

Study Design

Significance Level: 0.000000050

Disease Model: Additive

Prevalence: 0.0100

Disease Allele Frequency: 0.0900

Genotype Relative Risk: 2.4600

Results

Expected power for a one-stage study	1.000
Expected disease allele frequency	
Cases	0.185
Controls	0.088
Probability of disease	
Genotype A/A [with frequency 0.008]	0.03
Genotype A/B [with frequency 0.164]	0.0
Genotype B/B [with frequency 0.828]	0.0

BIN1 rs6733839

Cases/Controls = 0.028

Sample Size

Cases: 2843

Controls: 100000

Study Design

Significance Level: 0.000000050

Disease Model: Additive

Prevalence: 0.0100

Disease Allele Frequency: 0.3950

Genotype Relative Risk: 1.2500

Results

Expected power for a one-stage study	0.982
Expected disease allele frequency	
Cases	0.445
Controls	0.394
Probability of disease	
Genotype A/A [with frequency 0.156]	0.0
Genotype A/B [with frequency 0.478]	0.0
Genotype B/B [with frequency 0.366]	0.0

Some additional minor comments:

– I know there is no much room in the manuscript, but it would be important for the readers to know what do the 20% individuals with RBD who do not develop neurological disorders look like. For example, do they also have synuclein aggregates? Perhaps just a mention and cite a few articles if available.

These is a great point. In fact, the most recent and largest clinical study of iRBD, in which we were also involved, shows that it is over 80% that developed synucleinopathy after 15 years of follow-up. The rest were followed up for a shorter period or were lost to follow up / passed away. It is possible that if they had lived long enough and could be kept in the study, the % of convertors will be much higher. Unfortunately, there are no pathological studies of people who just had RBD and did not convert. There are however, imaging studies, showing that even in RBD patients who did not convert yet there are signs that can be identified in MRI indicating neurodegeneration similar to that seen in PD/DLB. For example, the study by Rolinski et al. (*Brain* 2016) showed basal ganglia connectivity dysfunction in RBD that is highly similar to that seen in PD. However, these individuals are not necessarily part of the <20% that did not convert, since they had RBD mostly for shorter period than 15 years.

Therefore, to address this point, the relevant part of the introduction now reads (page 4, lines 114-118):

“Over 80% of iRBD patients will convert within 10-15 years on average, to Parkinson’s disease (PD), dementia with Lewy bodies (DLB), or in rare cases, multiple system atrophy (MSA).^{2,3} It is still unclear whether the remaining iRBD patients who did not convert at long follow-up will eventually convert, and pathological and imaging studies of this specific population are warranted.”

– Just a formatting issue: Something is off with the period like in line 155, 12.3%, and the following standard error 0.07, the period is higher than it should be

Thank you for noticing, we fixed it.

– I wonder if the negative result with PRS and PD-RBD could be due to a lack of statistical power instead of tagging RBD-specific loci.

Thank you for this comment. We added this as an additional potential limitation in the discussion section, which now reads (page 14, lines 450-454):

“Power must be considered when interpreting these results, particularly in the lack of association of RBD PRS with PD without RBD, and in the genetic correlation analyses where we observe high

variability for some traits, however we find it important to report these results as foundational work in RBD genetics.”

– Studies in SNCA show there are at least 2, maybe even 3 independent signals coming from the gene and risk of PD. Authors should mention if rs3756059 is in LD with any of the 5’ SNPs associated to PD.

This is again a good point and we added it in the discussion, in pages 11-12, lines 380-382:

“This locus is in LD with a secondary PD GWAS variant, rs7681154, ($R^2=0.99$, $D'=0.97$), and is potentially a marker of the PD+RBD subgroup,³⁰ which is a group likely to develop PD with dementia.^{3”}

– As I mentioned at the top, one of my main issue with the manuscript is the lack of proof towards RBD being a distinct genetic entity. As an example, the statement in the discussion: “A distinct genetic background for RBD is also supported by the lack of genetic correlation between iRBD and the most recent PD GWAS...” is not justified as there is quite a lot of overlap between both as shown in Figure 5A, and it is only in a few loci that the direction is different between the two.

We agree with this comment, and we toned down the tone and modified the language in the relevant sections throughout the manuscript.

What we meant by distinct is that if, for example, you look at Manhattan plots of PD, DLB and RBD you can distinguish which is which. In DLB you will have APOE and BIN1 hits, which do not exist in PD and RBD, in PD you will have MAPT and LRRK2 hits that do not exist in DLB and RBD. So the overall genetic background in that sense seems to be distinguishable, yet we agree that the term “distinct” is too strong and we therefore modified it.

For example, we deleted the sentence “*A distinct genetic background for RBD is also supported by the lack of genetic correlation between iRBD and the most recent PD GWAS*”, changed sentences where the reader might get the wrong impression that the genetic background of RBD is completely distinct, and we use more terms like “partially overlap” when discussing the genetic backgrounds of RBD, PD and DLB, to emphasize that some of the genetic background is the same and some is different.

– The authors should mention if the check for any association with BST1, or LAMP3 in their GWAS as they recently published an association between RBD and variants in these 2 genes. I am surprised they did not even cite their work.

This was added now to the results, in page 6, lines 169-177, which now read:

“Additionally, we investigated common variants in loci where rare variants are associated with reduced RBD risk: BST1 and LAMP3.¹⁵ Intronic BST1 SNP rs4389574 (MAF=0.45) shows a potential protective effect in the RBD meta-analysis, without confidence (beta=-0.07, se=0.03m p=0.01). Intronic LAMP3 SNP rs3772714 (MAF=0.14) also shows a potentially protective effect, again without confidence at GWAS-corrected significance (beta=-0.14, se=0.04, p=0.001). Linkage disequilibrium (LD) is difficult to assess between common and rare variants; in this case, LDlink¹⁶ shows the minor alleles of the RBD rare variants in BST1 (rs6840615) and LAMP3 (rs56682988) correlate with the minor alleles for the common variants reported above, however in both cases only one instance of the rare variant was identified.”

– The authors should mention what the LD is between the two GBA variants in Table 1.

Thank you for this comment. We know that these variants are not in LD since they represent the E326K and N370S variants of GBA. We know from Gaucher’s disease that the vast majority of N370S carriers do not carry E326K, and vice versa. Furthermore, carriers of both variants have been reported in trans and not in cis. Calculating the accurate LD statistics is a bit challenging due to the rarity of the variants. To explain this, we added the following to the legend of Table 1:

“The two GBA associations, representing the p.Glu326Lys and p.Asn370Ser variants, are not in LD, as they are known from Gaucher’s disease studies to reside on different alleles.”

Reviewer #2

GWAS of RBD identifies novel loci with distinct polygenic and brain expression effects
The authors report the results of the first GWAS of RBD, with the identification of 6 independent associations in 5 genome-wide significant susceptibility loci, each labeled with the symbol of the nearest gene (SNCA, GBA, TMEM175, INPP5F and SCARB2). The authors also conducted post-GWAS (coloc, geneset enrichment, polygenic risk score, and genetic correlation) analyses and, based on their results, claim RBD is a genetically distinct alpha-synucleinopathy subtype. RBD is described as either 1) a clinical symptom that precedes diagnosis of alpha-synucleinopathies [mainly PD and DLB] by 10+ years [referred to as iRBD], or 2) a clinical symptom that rather frequently manifests itself after diagnosis of alpha-synucleinopathies [referred to as sRBD, or probable RBD (pRBD) when ascertained via a questionnaire]. The authors specifically describe RBD as a “more

malignant subtype of alpha-synucleinopathies” and iRBD as a “prodromal alpha-synucleinopathy”.

Specifically, the authors performed two GWASes, followed by meta-analysis:

1) A GWAS cohort [iRBD] with 1,061 iRBD cases and 8,386 controls (no iRBD)

2) A GWAS cohort [23andMe] with 1,782 PD+pRBD cases and 131,250 controls (no PD and no iRBD)

RBD ascertainment differs for the two cohorts (clinical diagnosis vs. self-reported based on single question).

Here are my comments related to each subsection of the Results section, followed by some minor issues and a general one, which - in my opinion - would need to be addressed prior to publication in Nature Communications.

Thank you for your thorough and overall positive assessment of our work.

GWAS

• For the ease of the reader, in Methods - GWAS, please add a brief description of how the 23andMe cohort was collected, genotyped and filtered in addition to citing previous work. More importantly, it is not clear how the 23andMe GWAS was performed. Were variants imputed and rvtests used as described for “All other cohorts”? If done differently please provide a detailed description of the procedures used. If NOT done differently, please make it clear in the text.

Thank you for this note, we have now added this information in the Methods section in page 17, lines 554-568:

“23andMe cohorts were collected, genotyped, and filtered as previously described.⁹ Briefly, all individuals included in the analyses provided informed consent and answered surveys online according to the 23andMe human subject protocol, which was reviewed and approved by Ethical & Independent Review Services, a private institutional review board (<http://www.eandireview.com>). DNA extraction and genotyping were performed on saliva samples by National Genetics Institute (NGI), a CLIA licensed clinical laboratory and a subsidiary of Laboratory Corporation of America. Samples were genotyped on one of five genotyping platforms. The v1 and v2 platforms were variants of the Illumina HumanHap550+ BeadChip, including about 25,000 custom SNPs selected by 23andMe, with a total of about 560,000 SNPs. The v3 platform was based on the Illumina OmniExpress+ BeadChip, with a total of about 950,000 SNPs. The v4 platform was a fully customized array, including about 570,000 SNPs. The v5 platform, in current use, is an Illumina Infinium Global Screening Array (~640,000 SNPs) supplemented with ~50,000 SNPs of custom content. Samples that failed to reach 98.5% call rate

were re-analyzed. Individuals whose analyses failed repeatedly were re-contacted by 23andMe customer service to provide additional samples.”

- It is mentioned that rare genotyped variants were included if previously shown to be associated with alpha-synucleinopathies. The full list of these variants and their QC parameters, in particular MAC in cases and controls, and differential missingness test statistics for each GWAS cohort, should be included in the supplementary materials.

This has been added to the supplementary (Supplementary table 1) as suggested. We also added a sentence in the manuscript to say that the RBD PRS remained stable without the only rare variant included in it (*GBA1* p.N370S) in page 6, lines 193-196:

“Results are comparable when excluding rare *GBA* variant p.Asn370Ser from the polygenic risk profile, with only the *iRBD* AUC decreasing from 0.61 to 0.60 (95% CI 0.56-0.64), but the difference between *iRBD* and *PD-pRBD* AUC becomes no longer significant ($p=0.06$).”

Supplementary Table 8

SNP	Gene	Protein change	% carriers, cases 23andMe/McGill	% carriers, controls 23andMe/McGill	GT rate 23andMe/McGill
rs76763715	GBA	p.N370S	0.020/0.013	0.007/0.003	>0.99/>0.99
rs34637584	LRRK2	p.G2019S	0.007/0.00	0.002/0.0006	>0.99/>0.99

- In addition to the above, the authors should also explain how error rates due to the large imbalance in cases and controls in both cohorts (but particularly severe for the 23andMe cohort) were controlled.

This is now added to the section about GWAS in the 23andMe data in the Methods section (page 17, lines 578-580):

“P-values were computed using a likelihood ratio test and adjusted for LD-score regression intercept to account for sample size mismatch between cases and controls.”

- One point highlighted by the authors in the title/abstract and main text is that RBD is genetically distinct from PD and DLB based, in part, on the lack of significant

associations at well-established PD and DLB GWAS loci in their RBD GWAS. However, as mentioned by the authors themselves, lack of statistical significance does not prove the null hypothesis and, in an admittedly small GWAS study as the one described by the authors, lack of significant association would indeed be a common occurrence even in the presence of a true effect. Indeed, this has been observed throughout the history of GWAS for several traits and diseases as sample sizes increased over time. In addition, since statistical power is mentioned by the authors repeatedly, it would be useful to include the results of power analyses of their RBD GWAS (iRBD, PD+pRBD, and meta) for MAFs and effect sizes commonly observed at PD and DLB GWAS loci.

Thank you for making this comment, which was also made by Reviewer 1. We agree that the word distinct might be confusing. As we explained in our response above, what we meant by it is that if one is looking at the Manhattan plots of PD, DLB and RBD, it is possible to distinct which is which. In PD there will be hits in the LRRK2 region for example, which do not appear in DLB and RBD (despite adequate power, see the figure on LRRK2 in the response to reviewer #1), in DLB there will be hits in APOE and BIN1 which do not appear in PD and RBD.

However, we agree of course that lack of statistical significance does not prove the null. Therefore throughout the manuscript we toned down the language and explained better what we mean given the data and given the limitations of statistical inference and power.

Regarding power, we have calculated the power for different allele frequencies for the meta analysis (it is not a common practice to calculate power for each cohort in a meta analysis, and the meta analysis results are those we mainly refer to throughout the paper). The figure below, where we calculated the power for different allele frequencies with different effect sizes has been added to the supplementary material.

• In support of their hypothesis that RBD is genetically distinct from PD and DLB, the authors also point to RBD association signals at two loci (SNCA and SCARB2) being independent from PD association signals at the same loci due to lack of LD between top SNPs. To further support this point, the results of conditional analyses at these two loci should be included in supplementary materials. I would also like to see a conditional analysis/comparison of RBD GWAS association signals (iRBD, PD+pRBD, and meta) with those in PD AAO GWAS loci discovered by the same authors, since RBD is mentioned to be associated with a more malignant subtype of alpha-synucleinopathies. Indeed, all 5 RBD loci are also PD AAO loci and the distinct genetic architecture of RBD susceptibility could be related to the one observed by the same authors between PD susceptibility and AAO. A conditional analysis/comparison with recently reported DLB GWAS loci should also be included, since iRBD can progress to both PD and DLB.

In the two loci mentioned by the reviewer, we have indeed performed a conditional analysis showing no residual associations after conditioning on the top hits.

Conditional analysis for SCARB2, showing lack of association of the top PD risk SNP before and after conditional analysis:

For the conditional analysis of *SNCA*, this has already been done extensively by us in a paper focused on the *SNCA* locus in RBD (Krohn et al. 2020 *Annals of Neurology*). Repeating it here will be redundant (identical results), therefore we have referred in the text to this paper and its main conclusions regarding the conditional analysis, in page 5, lines 158-163:

“These five loci have also been implicated in PD,¹³ however the RBD-associated SNPs in SNCA and SCARB2 are not in LD with the top PD-associated SNPs in these loci, and are thus considered independent. No secondary associations were identified by conditional-joint analysis; notably, the PD variant is not significant at the SCARB2 locus (Supplementary Figure 3). The SNCA locus

structure in RBD has been extensively studied before, with a potential secondary hit that is below genome-wide statistical significance.⁹

As for the AAO, the SNPs associated with PD AAO are the same SNPs associated with PD risk (although in SCARB2 and INPP5F they are not statistically significant at the GWAS level), therefore it would be the same conditional analysis.

For the DLB loci, we have mentioned in the text (page 5, lines 163-168) that the specific loci associated with DLB (as opposed to PD), *BIN1* and *APOE*, are not associated with RBD:

“Additionally, PD or DLB-associated SNPs in notable GWAS loci, such as MAPT (rs62053943), LRRK2 (rs34637584), BIN1 (rs6733839) and APOE (rs769449)^{8,14} are not associated with RBD at this sample size, which had sufficient power (>80%) to detect the effect sizes seen in PD and DLB.”

• The authors should also perform a GWAS comparing PD cases with and without pRBD in the 23andMe cohort, in order to identify loci specific to RBD susceptibility in PD. The authors should also consider performing a GWAS based on multinomial logistic regression to include all classes (PD+pRBD, PD-pRBD, and controls, and possibly also iRBD) and thus increase power and more rigorously identify loci with shared or distinct effects across different disease subtypes (see for example <https://doi.org/10.1093/bioinformatics/btw075> and <https://doi.org/10.1002/gepi.20486>).

We appreciate this suggestion by the reviewer, as it is the most logical next step. We have applied to 23andMe to get the data, which is a long process, and if approved (which could take at least 6 months), this would be done in about a year the earliest. Furthermore, we do not expect that the results will be the same. Preliminary analysis of our own data shows that LRRK2 for example comes up in this analysis, as it is associated with PD without RBD and not with PD with RBD. We see this study as a follow-up, separate study, which will only be completed once we get all the data from 23andMe and perform full analysis, and therefore outside of the scope of the current paper.

• One major limitation of this GWAS is lack of replication in an independent cohort. However, an independent cohort including PD+pRBD (N=285), PD-pRBD (N=217) and controls (N=900 or 907 is reported at two different places in the text, this discrepancy should be resolved) was used for PRS studies. Therefore, replication (and subsequent meta-analysis) of genome-wide and suggestive associations identified by the PD+pRBD vs controls GWAS (performed by the authors) and the PD+pRBD vs PD-pRBD GWAS (suggested above) is possible and should be performed.

First, thank you for noticing the N error, it is 907 and we changed it in the text.

We thank you for this suggestion, and while this is a very small cohort even for replication, we performed the analysis, and included the results in Supplementary Table 2, attached below as well. As can be seen, 3 associations are replicated at a nominal level, one (*SCARB2*) has the same direction of effect with $p=0.09$, and also one GBA variant has the same direction of effect but far from statistical significance, probably due to reduced power (28% for this variant at this sample size, power for each variant is also included in the table).

Supplementary Table 7, comparing PD+pRBD (N=285) and controls (N=907).

SNP	Position (hg19)	Closes gene	EA	Odds ratio (95% CI)	p	Power
rs3756059*	4:90757272	SNCA	A	1.22 (1.00-1.48)	0.048	0.70
rs12752133*	1:155205378	GBA	T	3.00 (1.02-8.83)	0.047	0.40
rs76763715	1:155205634	GBA	C	1.30 (0.05-33.04)	0.888	0.28
rs34311866*	4:951947	TMEM175	C	1.28 (1.62-2.04)	0.041	0.40
rs117896735	10:121536327	INPP5F	A	0.94 (0.43-2.05)	0.887	0.37
rs7697073	4:77132634	SCARB2	T	1.19 (0.97-1.46)	0.090	0.39

PRS

● **This section should be revised in light of results from the PD+pRBD vs PD-pRBD or multinomial GWAS mentioned earlier. The ability of the RBD PRS to differentiate between PD+pRBD and PD-pRBD cases should also be assessed. Indeed, the title of this section points to the distinction between iRBD (or PD+pRBD) and PD-pRBD, yet (if I understood correctly, given that the predicted outcomes are clearly indicated only for the iRBD AUC) the AUCs reported in the main text are between iRBD (or PD+pRBD or PD-pRBD) and controls.**

We have now added this to the PRS section as follows, in page 6, lines 189-192:

“When comparing PD+pRBD to PD-pRBD, the RBD PRS is not a strong predictor (AUC=0.55, 95% CI 0.52-0.59). However, while we argue this RBD PRS is enriched for RBD loci, it does not capture other key differences we may expect to see in PD +/- RBD, such as LRRK2 and MAPT variants.”

- **Is the RBD PRS associated with PD AAO?**

Our analysis did not show association with PD AAO, and we added this to the Results as well, in page 6, lines 204-205:

“RBD PRS was not associated with changes in RBD AAO, PD AAO, or rate of conversion from RBD to overt neurodegeneration.”

DGE

- **This section focuses on the comparison between the sizes, direction and p-values of eQTL effects of RBD and PD-associated variants in the SNCA and SCARB2 loci across several brain regions. The stated purpose of this analysis is to test the hypothesis that the different disease-associated SNPs have heterogeneous effects on regional gene expression in the brain. First, I find the titles of this section and of Fig. 2 a bit misleading, as they seem to refer to differential gene expression (DGE) rather than differential eQTL effects across brain regions. Indeed what’s shown in the figure and presented in the text are eQTL effects of SNPs, not differential gene expression of SNCA and SCARB2 in different brain regions. Please clarify this in both titles and change “GTEX v8 Expression” in Fig. 2 to “GTEX v8 eQTL effects”. Second, the only pairwise eQTL effects that survive multiple testing correction are observed for SNCA-AS1 (this should be noted in the text, in particular in the Discussion when claiming rs7697073 is associated with SCARB2 expression). While it is true that more statistically significant SNCA-AS1 eQTL associations are observed for the RBD variant compared to the PD variant, from this visual comparison it is impossible to assess the statistical significance (or simply the actual significance) of this difference, given that the direction of effect is consistent across the RBD and PD variants and no permutation (or otherwise valid NHST) p-values are provided for the actual hypothesis. This should be resolved, since the results of this analysis are one the main conclusions highlighted in title of the manuscript.**

Thank you for these comments, we agree.

We changed the titles of this section and Fig 2 to better reflect our findings.

As suggested, we also toned down the discussion part touching *SCARB2* expression, as we agree that these results are currently preliminary and suggestive, and should be studied further in follow up studies. The relevant part of the discussion now reads (page 12, lines 387-398):

“We found a similar phenomenon in the SCARB2 locus; rs7697073 is associated with RBD in the current GWAS, whereas in the recent PD GWAS there is an independent association at rs6825004. The PD-associated variant is possibly associated with SCARB2 expression in the substantia nigra, while the RBD-associated variant, rs7697073, is not. This potential difference in SCARB2 expression should be considered with caution, as the association with expression in the substantia

nigra does not survive correction for multiple comparisons. We can hypothesize that if this difference will be proven to be true, it may lead to an earlier degeneration of the nigrostriatal fibers in the PD cohort compared to the RBD cohort, thus explaining the earlier manifestation of motor symptoms in the former. In RBD, the top associated variant in this locus (rs7697073), like the SNCA locus variant, is potentially associated more with expression in cortical brain regions, providing additional support for our hypothesis, although here too this association does not survive Bonferroni correction.”

Regarding the statistical testing suggested, this is a great point and suggestion, unfortunately we cannot compare the two statistically because we cannot access the individual level data in GTEx. Therefore, we added it as an additional limitation in page 8, lines 261-266):

“Although the observed direction of eQTL effect on SNCA-AS1 expression is consistent between the RBD and PD variants, only the RBD variant is causally linked to SNCA-AS1 expression via colocalization. The differential strength and patterns across the brain regions between the PD and RBD variants are an intriguing field for follow-up investigation; in individual GTEx data, statistically comparing the mRNA levels data associated with the PD or RBD variant in each tissue could help clarify this observation.”

• The authors mentioned using multiple testing correction as done by GTEx; however, this should be better described in the manuscript since multiple testing correction depends on the hypothesis being tested and GTEx procedure may not be relevant here.

We agree and now explain this point better in the text, in page 8, lines 252-254:

“A variant is considered associated with expression in a tissue based on its association with mRNA levels after FDR correction by GTEx, however in our case it does not indicate that this is the causal eQTL; it may be in LD with the causal SNP.”

• If the authors firmly believe that this visual presentation is critical to support the main conclusion of the manuscript, then the same comparisons should be made between RBD [and iRBD] and DLB-associated variants, and ideally also between RBD [and PD+pRBD] and PD:AAO-associated variants. This would substantiate the authors claim

that RBD is distinct from PD and DLB, and begin to investigate the possibility that RBD vs PD distinction may (or may not) be largely due to a faster progression rate in PD+pRDB vs PD-pRDB.

This is a good point. First, we would like to clarify that we present these variants because they are distinctly different in PD vs the current RBD GWAS. They have a different direction of association and different, independent SNPs associated with each trait in these loci.

In DLB, the loci identified have either the same SNPs with the same effect (therefore there is no point analyzing them in the same way – it is the same SNPs or SNPs in strong LD), or loci that are not associated with RBD at all (APOE and BIN1), and therefore we do not have SNPs in RBD from these loci to compare.

Regarding AAO – these are again the same SNPs (or SNPs in strong LD) that affect the risk of PD and AAO of PD, therefore the analysis will be the same as we have already done for PD risk.

• I would also consider rewording the rather confusing title of this section because it is unclear to me how the analyses presented in this section relates to the fact that “differential gene expression in different brain regions may drive the independent associations of SNCA and SCARB2 [locus SNPs] with RBD and PD”. If by “driving” the authors mean “mediating” then conditional SMR and/or PrediXcan-like analyses should be performed instead.

We agree and we reworded it to:

“Differential eQTL effects in different brain regions may shed light the independent associations of SNCA in RBD and PD”

Coloc

• Also in this set of analyses, evidence of colocalization is observed only at the SNCA locus but not at the SCARB2 locus. This should be noted in the text, in particular in the Discussion when claiming rs7697073 may be acting on disease risk via modulation of SCARB2 expression.

This is another good point, and this is mentioned now in the Results (page 8, lines 271-274):

“These SCARB2 results must be taken with caution and examined further, as the eQTL associations are not statistically significant after multiple testing correction, and we did not find evidence of colocalization in the RBD SCARB2 locus in brain or blood (Supplementary Figures 7-9).”

And in the Discussion, where we deleted the words “and *SCARB2*” in the first paragraph of the discussion, and also toned down the *SCARB2* discussion as detailed in a previous comment.

• **Given the interesting association between SNCA locus SNPs and MMRN1 expression in the blood and the specificity of MMRN1 expression in microglia of the brain, it would be interesting to perform colocalization analyses using myeloid and/or microglia-specific eQTL datasets that are publicly available (see for example <https://doi.org/10.1038/s41588-021-00976-y>).**

We performed these analyses and suggested and did not find statistically significant associations. This is now added in the text-

In the methods (page 20, lines 658-660):

“We additionally investigated eQTLs from individual CNS cell types (astrocytes, endothelial, microglia, oligodendrocytes and precursors, and pericytes) using eQTLs generated by Bryois et. al.”

In the results (page 7, lines 232-234):

“We further investigated whether these associations in CNS cell-type specific eQTLs, but did not find evidence of colocalization with any RBD loci.”

Pathway analysis

• **It is unclear how the list of candidate causal genes for pathway analysis was generated and what genes were included in it. In the Methods section the authors state: “RBD genes included were those closest to the most significant GWAS SNP” which is not a very precise description of the selection process that was used. Does this mean that genes within a certain distance range from a single SNP (the most significant GWAS SNP) were selected? Or (most likely) that, for each locus, only one gene (the one closest to the lead SNP in each locus) was selected? Were only genome-wide significant loci (hence only 5 genes) considered or also loci with suggestive evidence of association? More details are needed to better understand what was actually done.**

Thank you for the opportunity to clarify this. Indeed, we selected one gene per locus for analysis. Choosing multiple genes from each locus can create false associations, since it is possible that some genes from the same locus are involved in the same pathway or cellular process or

compartment, which may result in false enrichment. We now clarify this better in the text, in page 19, lines 641-643:

“RBD genes included were those closest to the most significant GWAS SNP at the GWAS significance level, a single gene from each locus, as including multiple genes from the same locus may lead to false enrichment.”

• The authors should also justify why the nearest-gene criterion was utilized rather than, for example, nominating genes based on eQTL effects, or including all genes within or near the association region, for example. The example of BAG3 highlighted by the authors should, by itself, justify trying a different approach in addition to nearest-gene criterion.

The nearest gene is most straight forward way to perform this analysis, and in most cases, the nearest gene is the associated gene. For example in PD – SNCA, LRRK2, GBA, TMEM175, VPS13C, GCH1, all are GWAS loci where the gene is known or have very strong evidence for it, and in all of them the gene is the nearest gene.

However, in many loci, there are QTLs for multiple genes. For example, if we take the *GBA* associated SNP (rs12752133), it has eQTLs in GTEx with *GBA*, *GBAP1*, *HCN3* in different tissues, it has sQTLs with *FDPS*, *TRIM46*, *MSTO1* *YYIAP1*, *SCAMP3* and *SLC50A1*. And this is only in the tissues in GTEx. There are many other resources, including single cell data, where it is likely that QTLs exist with other genes in the *GBA* locus. This makes pathway analysis based on QTL data not feasible, as many combinations of genes are needed for the analysis, as each time only one gene per locus should be included. We therefore opted for the analysis of the nearest gene. Of course, there are cases where it will not necessarily be the nearest gene, but since this is an analysis that in the end just suggests potential pathways, that require further studies, we think this is currently the best approach.

We have now better justified it in the text, which now reads (page 19, lines 643-645):

“We opted for choosing the nearest gene and not using QTLs, since in many loci there are multiple QTLs in multiple tissues with multiple genes, which will make the selection of genes for this preliminary analysis based on QTLs challenging.”

• The supplementary table should also include the ID/Symbols of the genes that belong to each geneset or at least which of the GWAS-nominated genes are part of each geneset, in addition to the geneset size.

We added the GWAS nominated genes that are part of each enriched gene-set as suggested in the supplementary table.

Comparison of PD with DRB GWAS loci

• **Please add a panel to Fig. 5 to show the “PD vs RBD” (i.e., meta of iRBD and PD+pRBD) comparison.**

This has been added.

• **Are all the DLB GWAS-significant variants also included in this list of PD GWAS loci? Please generate additional figure and supplementary table like the ones shown, but using DLB instead of PD GWAS sumstats.**

We now added variants that are unique to the DLB GWAS (i.e. not already represented by the PD GWAS variants). We also changed the figure so that it shows, according to the shape of the variant, for how many GWASs this specific variant is associated with (including the PD, PD AAO, DLB and RBD GWASs).

We also added a supplementary figure to show these same variants, but instead of comparing to PD case-control GWAS betas, we compare with PD AAO summary statistics. As expected, most loci that increase risk for any of the synucleinopathies are associated with an earlier AAO, so we see a consistent “negative” effect direction. However, in iRBD, we see a few with the opposite effect; for example, the PD *SCARB2* is nominally associated with decreased risk for RBD and an earlier AAO for PD. The same locus that shows a differential effect between PD and the RBD meta-analysis, *CYLD*, shows this same pattern for AAO in iRBD.

Heritability and genetic correlation

- **The lack of sufficient power to accurately measure genetic correlation between iRBD and DLB is mentioned. Please describe how power was calculated and what its value was.**

This is now explained in the results section “LD score regression reflects potential differences between iRBD and PD+pRBD” in page 10, lines 321-325:

“Therefore, this possible association is likely driven by iRBD, although we could not accurately measure genetic correlation between iRBD and DLB due to high variability causing the correlation estimate to be out of bounds ($r_g=1.28$, $se=0.56$, $p=0.009$) suggesting we are underpowered to detect a true effect.”

• Please discuss how this issue of limited power of iRBD GWAS to measure genetic correlations with other traits may confound the interpretation of the LD Score results as providing evidence for a distinct genetic architecture of iRBD.

We added this to the limitations section in the discussion (page 14, lines 450-454):

“Power must be considered when interpreting these results, particularly in the lack of association of RBD PRS with PD without RBD, and in the genetic correlation analyses where we observe high variability for some traits. This issue may confound the LD score regression results; however we find it important to report these results as foundational, preliminary work in RBD genetics.”

• Please correct the claim that iRBD is genetically correlated with type II diabetes, since the correlation is not significant after multiple testing correction.
• In light of the above it is surprising that the positive (and almost significant after multiple testing correction) correlation between RBD and Alzheimer’s disease was not discussed.

We made the change to the type II diabetes claim as suggested, and touched on the Alzheimer’s association at the end of “LD score regression reflects potential differences between iRBD and PD+pRBD” results, in page 10, lines 334-338:

“Interestingly, PD+pRBD may be correlated with Alzheimer’s disease ($r_g=0.30$, $se=0.15$, $p=0.04$), again without confidence, which we do not see in iRBD at this sample size, although the correlation coefficients are similar ($r_g=0.33$, $se=0.20$, $p=0.11$). Those with Type II Diabetes are at increased risk for Alzheimer’s²⁵ and the two conditions share genetic risk architecture.”

• Given that, in previous sections, the authors claimed distinct genetic architecture of RBD compared to PD based on distinct association signals at a couple of loci, and yet the results of these global genetic correlation studies are inconclusive and at the same time unsurprising, the authors’ claim would be strengthened by performing local/locus-specific genetic correlation analyses between RBD (iRBD, PD+pRBD, and meta) and PD/DLB at those loci.

Thank you for this suggestion. We now also pruned the summary statistics to just include PD, RBD, and DLB hits +/- 500kb, and then we re-did the genetic correlation and added results to this section, which now reads (page 10, lines 327-332):

“This difference between iRBD and PD+pRBD is quite pronounced when examining genetic correlation between the synucleinopathies for only PD, DLB, and RBD GWAS loci +/- 500kb. PD+pRBD is correlated with PD ($r_g=0.76$, $se=0.08$, $p=1.2E-19$) with uncertain results for DLB ($r_g=1.31$, $se=2.03$, $p=0.52$), and iRBD is not correlated with PD ($r_g=0.19$, $se=0.13$, $p=0.15$) or DLB, although with a potentially high correlation coefficient with DLB ($r_g=0.91$, $se=0.56$, $p=0.17$) with low confidence.”

Minor issues

● **In Methods - Population, please report average age + standard deviation of 23andMe cohort, separately for cases and controls.**

This is now added.

● **Describe how individuals in both cohorts were confirmed of European ancestry and ancestry outliers/mixed ancestry individuals removed using HapMap 3 as reference (visual inspection of PC plot? distance from ref pop centroid? global ancestry estimation?). If visual inspection was used, please include PC plot(s) in supplementary materials.**

This is now explained this in methods, in page 17, lines 569-575:

“Participants were restricted to those of European ancestry determined through an analysis of local ancestry.⁵⁹ Briefly, a support vector machine (SVM) is used to classify individual haplotypes into one of 31 reference populations (<https://www.23andme.com/ancestry-composition-guide/>). The SVM classifications are then fed into a hidden Markov model (HMM) that accounts for switch errors and incorrect assignments, and gives probabilities for each reference population in each window. Finally, simulated admixed individuals are used to recalibrate the HMM probabilities so that the reported assignments are consistent with the simulated admixture proportions.”

- QQ plot of individual RBD GWASes and meta-analysis should be included in supplementary materials.

This is added as Supplementary Figure 1 as suggested.

- The locus labels are very difficult to read in Fig. 1A, please consider changing the color scheme.

We agree and made the change:

- For the ease of the reader, please include a table where the sumstats across all disease traits of all RBD, iRBD, PD+pRBD, PD, PD:AAO and DLB-associated variants discussed in the main text are presented together.

We now added this to the supplementary table with PD loci (newly Supplementary Table 5).

General issue

While the authors do a good job of highlighting the limitations of their study (e.g., limited size) and briefly point to lack of statistical significance as unable to prove absence of an effect, often their claims in the discussion may still lead the reader down the path of the most common mis-interpretation of non-significant p-values in the biomedical literature, i.e., as evidence of a lack of effect. The authors should revise their writing in order to avoid this confusion and should also include effect size estimates and confidence intervals whenever reporting or citing an effect in their manuscript. Alternatively, the authors could use statistical methods other than NHST (e.g., Bayesian inference) to calculate $P(H_0 | \text{data})$ under a clearly stated set of assumptions.

We agree with this comment, and made the relevant changes in the discussion to make things clearer and tone it down to avoid misinterpretation.

REVIEWERS' COMMENTS

Reviewer #1 (Remarks to the Author):

The authors have addressed all my concerns and if the other reviewer agrees, I believe this manuscript is ready for publication.

Reviewer #2 (Remarks to the Author):

The authors have satisfactorily addressed my main concerns and the manuscript is now fit for publication in Nature Communications.